# EZH2-mediated PP2A inactivation confers resistance to HER2-targeted breast cancer therapy

Yi Bao[1,2], Gokce Oguz[2], Wee Chyan Lee[2], Puay Leng Lee[2], Kakaly Ghosh[2], Jiayao Li[3], Panpan Wang[3], Peter E. Lobie[1,4], Sidse Ehmsen [5], Henrik J. Ditzel [5,6], Andrea Wong[7], Ern Yu Tan[8], Soo Chin Lee[1,7 ✉] & Qiang Yu [2,9,10 ✉]

HER2-targeted therapy has yielded a significant clinical benefit in patients with HER2+ breast cancer, yet disease relapse due to intrinsic or acquired resistance remains a significant challenge in the clinic. Here, we show that the protein phosphatase 2A (PP2A) regulatory subunit *PPP2R2B* is a crucial determinant of anti-HER2 response. *PPP2R2B* is downregulated in a substantial subset of HER2+ breast cancers, which correlates with poor clinical outcome and resistance to HER2-targeted therapies. EZH2-mediated histone modification accounts for the *PPP2R2B* downregulation, resulting in sustained phosphorylation of PP2A targets p70S6K and 4EBP1 which leads to resistance to inhibition by anti-HER2 treatments. Genetic depletion or inhibition of EZH2 by a clinically-available EZH2 inhibitor restores *PPP2R2B* expression, abolishes the residual phosphorylation of p70S6K and 4EBP1, and resensitizes HER2+ breast cancer cells to anti-HER2 treatments both in vitro and in vivo. Furthermore, the same epigenetic mechanism also contributes to the development of acquired resistance through clonal selection. These findings identify EZH2-dependent *PPP2R2B* suppression as an epigenetic control of anti-HER2 resistance, potentially providing an opportunity to mitigate anti-HER2 resistance with EZH2 inhibitors.

[1] Cancer Science Institute of Singapore, Yong Loo Lin School of Medicine, National University of Singapore, Singapore 117597, Singapore. [2] Cancer Precision Medicine, Genome Institute of Singapore, Agency for Science, Technology, and Research, Biopolis, Singapore 138672, Singapore. [3] Cancer Research Institute, Jinan University, Guangzhou, China. [4] Tsinghua-Berkeley Shenzhen Institute, Guangdong Province and Shenzhen Bay Laboratory, Tsinghua University, Shenzhen, Guangdong Province, China. [5] Department of Oncology, Odense University Hospital, Institute of Clinical Research, University of Southern Denmark, 5230 Odense, Denmark. [6] Department of Cancer and Inflammation Research, Institute of Molecular Medicine, University of Southern Denmark, 5230 Odense, Denmark. [7] Department of Haematology-Oncology, National University Cancer Institute, National University Health System, Singapore 119047, Singapore. [8] Department of General Surgery, Tan Tock Seng Hospital, Singapore, Singapore. [9] Department of Physiology, Yong Loo Lin School of Medicine, National University of Singapore, Singapore 117597, Singapore. [10] Cancer and Stem Cell Biology, DUKE-NUS Graduate Medical School of Singapore, Singapore 169857, Singapore. ✉email: csilsc@nus.edu.sg; yuq@gis.a-star.edu.sg

HER2-targeted therapy had significantly improved the prognosis of patients with HER2+ breast cancer[1,2]. However, despite this, nearly half of such patients still suffer disease relapse due to intrinsic or acquired resistance to the treatments[3,4], demonstrating the failure of stratifying breast cancer patients who will benefit from HER2-targeted therapy based solely on HER2 status. Despite significant efforts to resolve this clinical problem, developing clinical treatments with additional targeting strategies to overcome resistance remains a pressing clinical problem. The lack of a complete understanding of the molecular processes that determine an effective outcome of therapy has hindered the development of a rational combination treatment from overcoming resistance to anti-HER2 therapies.

Aberrant activation of PI3K/AKT/mTOR or MAPK pathway downstream of HER2 is linked to resistance to HER2-targeted therapy. Dysregulation of the key components in the pathway, such as *PIK3CA* mutations or loss of *PTEN*, is connected to anti-HER2 resistance[5–9]. Moreover, the perturbation on many other molecules associated with treatment resistance, such as p95-HER2[10], MUC4[11], IGF-1R[12], and c-SRC[13], etc., relies on activating this pathway to exert the resistance. However, inhibition of the PI3K/mTOR pathway with everolimus, an mTOR inhibitor, showed limited improvement from trastuzumab-containing regimens, as shown in two phase III clinical trials[14,15]. Limited efficacy of mTOR inhibitors has been attributed to the release of negative feedback of mTOR on receptor tyrosine kinases (RTKs) and AKT[16–18]. The same feedback activation is also observed by AKT inhibitors[19], implying that PI3K or AKT inhibitors might also fail to alleviate resistance to HER2-targeted therapies. Therefore, although activated PI3K/AKT/mTOR signaling has been implicated in anti-HER2 resistance, a more effective therapeutic strategy targeting the pathway is needed to overcome the resistance.

Serine/threonine phosphatase PP2A is a tumor suppressor family often inactivated in human cancers[20,21]. PP2A negatively regulates numerous oncogenic pathways in tumorigenesis, such as Myc, Wnt, PI3K/AKT[22–24], and can directly dephosphorylate AKT and mTOR targets ribosomal protein S6 kinase beta-1 (p70S6K) and eukaryotic translation initiation factor 4E-binding protein 1 (4EBP1)[25–30], to maintain the equilibrium of phosphorylation activity of the PI3K/AKT/mTOR pathway. Therefore, it is unsurprising that PP2A activity has been linked to sensitivity to kinase inhibitors in cancer, including mTOR or MEK inhibitors[24,31]. The PP2A family functions as a heterotrimeric complex that contains one scaffolding A-subunit, one catalytic C-subunit, and one regulatory B-subunit, encoded by a wide array of genes[32]. The scaffolding subunit, PPP2R1A, often carries loss-of-function mutations in certain cancers[31,33], while other regulatory PP2A subunits have been found to carry deletions[34] or epigenetic repression[24,35]. To date, the role of the PP2A family in breast cancer oncogenesis and in modulating the response to anti-HER2 therapies has yet to be characterized.

## Results

***PPP2R2B* downregulation confers risk of disease progression and poor treatment response to anti-HER2 therapy.** To investigate the role of PP2A in breast cancer, we used the public database KM Plotter (http://kmplot.com/analysis/) to evaluate the prognostic values of 18 known PP2A subunits[36] in breast cancer. Analysis of relapse-free survival (RFS) identified seven PP2A subunits (*PPP2R2D, PPP2R2B, PPP2R5D, STRN, PPP2R2A, PPP2R5A*, and *PPP2R1B*), where low expression at lower levels signified a poor prognosis (Fig. 1a and Supplement Table 1). Among them, a few have been previously implicated

in modulating drug sensitivity in other cancer types, including *PPP2R2B, PPP2R2A, PPP2R5A*, and *PPP2R1A*[24,31,37].

Next, we used another public database, GOBO (http://co.bmc.lu.se/gobo/gsa.pl), to determine the prognostic value of the above seven-candidate PP2A subunits in different breast cancer subtypes. Interestingly, *PPP2R2B* was found to be the most significant ($P < 0.01$) in its association with distant metastasis-free survival (DMFS) in HER2+ breast cancer, but not in basal-like, luminal A, or luminal B breast cancer subtypes (Fig. 1b, c). In addition to DMFS, *PPP2R2B* downregulation, among the seven candidates, is also associated with poor RFS (Supplementary Fig. 1a), as well as poor overall survival (OS) in HER2+ breast cancer patients (Supplementary Fig. 1b). In both public database and our in-house patient specimens, *PPP2R2B* downregulation occurred in about 50% of HER2+ breast cancers compared to normal tissues (Supplementary Fig. 1c, d). These findings suggest that *PPP2R2B* downregulation occurs in a substantial proportion of HER2+ breast cancers, and is associated with a higher risk of disease progression.

Currently, most HER2+ breast cancer patients are treated with anti-HER2 therapies, and because a substantial fraction of treated patients later suffers distant relapse relapses due to therapy resistance, we sought to determine the relationship between *PPP2R2B* expression and therapeutic outcome. We analyzed a retrospective cohort of 16 treatment-naïve primary HER2+ breast cancers and seven HER2+ that have lesions from patients who relapsed with distant metastases following trastuzumab treatment. *PPP2R2B* expression showed marked downregulation in the relapsed metastatic tumors compared to the primary tumors ($P = 0.0035$; Fig. 1d), suggesting a correlation between *PPP2R2B* downregulation and trastuzumab resistance.

To evaluate the value of *PPP2R2B* expression in predicting response to HER2-targeted therapy, we took advantage of a phase II clinical trial patient cohort comprises of non-metastatic HER2+ patients treated with neoadjuvant lapatinib, an HER2 kinase inhibitor, plus chemotherapy (paclitaxel and carboplatin) (ClinicalTrials.gov; Identifier: NCT01309607). We found that tumors with lower levels of *PPP2R2B* expression showed significantly less size reduction after two cycles of neoadjuvant treatment compared to tumors with higher expression of *PPP2R2B* ($P = 0.0080$; Fig. 1e), supporting the notion of *PPP2R2B* downregulation being associated with poor tumor response to anti-HER2 treatment. Furthermore, in an online dataset of HER2+ breast cancer patients treated with a neoadjuvant trastuzumab-containing regimen (trastuzumab + paclitaxel) (GSE62327; $n = 24$)[38], six tumors with pathologic complete response (pCR) all showed higher *PPP2R2B* expression (Z-score $-0.5$ as cutoff) compared to tumors with an incomplete response (RD) where half of them showed lower expression of *PPP2R2B* (Fig. 1f). This finding was further supported by analysis performed on the ROC Plotter[39] (http://www.rocplot.org/) where lower *PPP2R2B* expression (Z-score $-0.5$ as the cutoff) was significantly associated with incomplete pathologic response ($n = 134$; $P = 0.0159$; Supplementary Fig. 1e) to trastuzumab treatment, and disease relapse in five years ($n = 44$; $P = 0.0457$; Supplementary Fig. 1f). Collectively, these clinical data established a strong association of *PPP2R2B* expression with clinical outcome of HER2+ breast cancer and suggested a potential role for *PPP2R2B* in determining the sensitivity of anti-HER2 therapies.

***PPP2R2B* downregulation is associated with intrinsic resistance to HER2-targeted therapies.** The above findings suggest that *PPP2R2B* expression might be a molecular determinant of sensitivity to HER2-targeted therapy; therefore, we sought to validate the functional role of *PPP2R2B* in anti-HER2 therapeutic

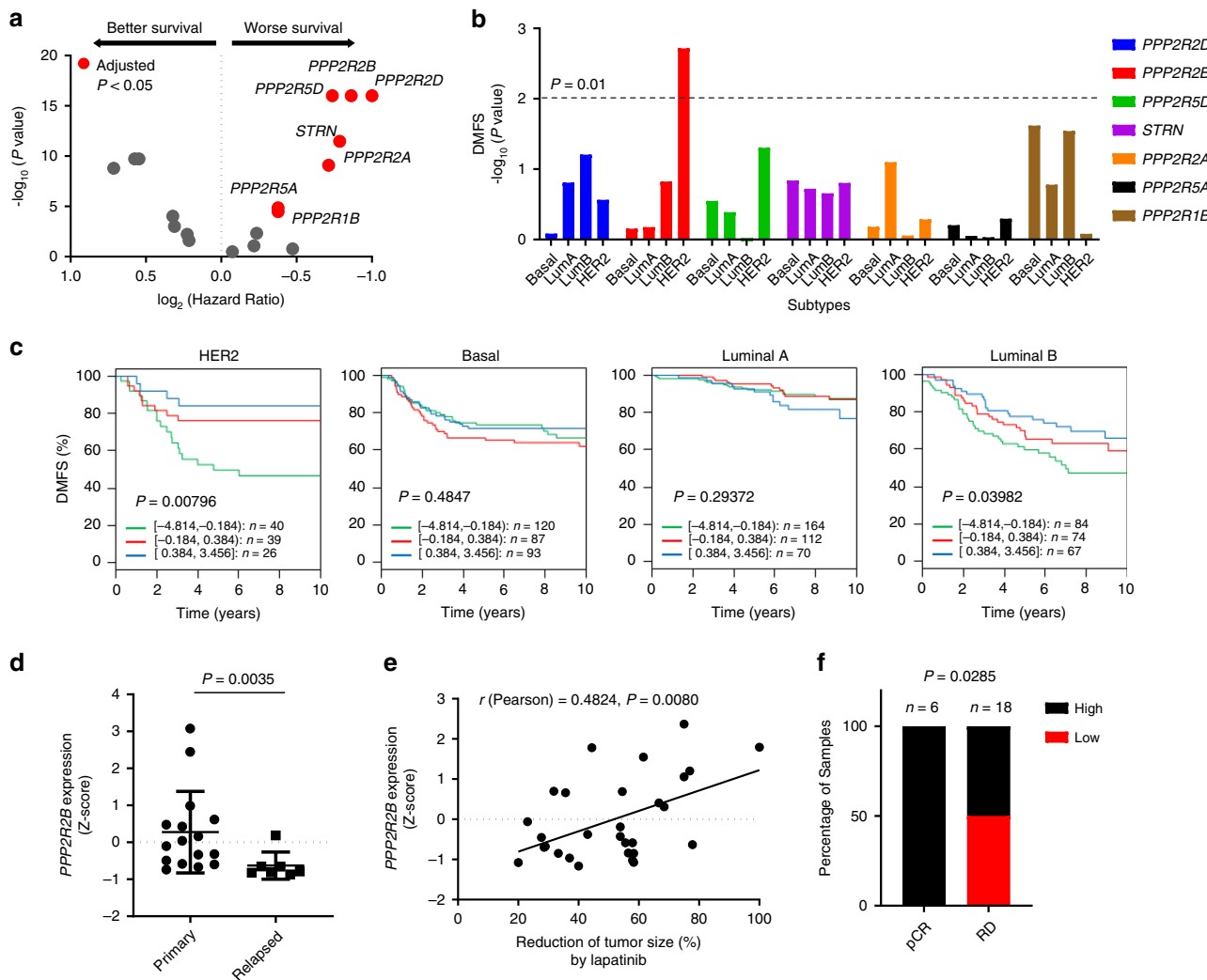

**Fig. 1 PPP2R2B downregulation confers risk of disease progression and poor treatment response to anti-HER2 therapy. a** Volcano plot illustrating hazard ratio ($\log_2$) compared with P value ($-\log_{10}$) between expression of PP2A members and relapse-free survival (RFS) in breast cancer. Members whose downregulation is significantly (Bonferroni-adjusted P value < 0.05) prognostic are highlighted in red, and their gene names are indicated next to their representative dots. Data were collected from KM Plotter (http://kmplot.com/analysis/). The computed based performing thresholds which are used as cut-offs were auto-selected for the percentiles of the subjects between the low and high gene-expression groups. **b** P value ($-\log_{10}$) between expression of the indicated subunit genes and distant metastasis-free survival (DMFS) in different breast cancer molecular subtypes. The molecular subtypes presented include basal-like (Basal), luminal A (LumA), luminal B (LumB), and HER2+ (HER2). P < 0.01 is considered statistically significant. **c** Kaplan–Meier curves comparing DMFS of breast cancer patients with tumors expressing the indicated levels of PPP2R2B in the indicated breast cancer subtypes. Data from **b** and **c** were collected from the GOBO database (http://co.bmc.lu.se/gobo/gsa.pl). P values in **a** to **c** were calculated with two-sided log-rank test. **d** RT-qPCR measuring expression of PPP2R2B in treatment-naïve primary HER2 + breast cancer tumors (n = 16) vs. distant metastatic tumors that recurred after a trastuzumab-containing regimen (n = 7). Data are expressed as means ± s.d. P value was calculated with two-sided Mann–Whitney U test. **e** Correlation analysis (n = 29) of PPP2R2B expression assessed by RT-qPCR and reduction of tumor size after two cycles of lapatinib-containing regimen in a HER2+ breast cancer cohort from a phase II clinical trial (ClinicalTrials.gov; Identifier: NCT01309607). The qPCR was performed with pre-amplified cDNA products. Correlation coefficient r and P value were calculated with Pearson correlation test. **f** Two-sided Chi-Square test of the relationship between low expression of PPP2R2B and residual disease (RD) after a neoadjuvant trastuzumab-containing regimen, in a HER2+ breast cancer cohort. pCR: pathologic complete response. High or Low PPP2R2B expression is defined with a cut-off of a Z-score of −0.5. Data are from microarray dataset GSE62327 available on Gene Expression Omnibus database (https://www.ncbi.nlm.nih.gov/geo/).

response. Alterations in the PI3K signaling pathway, including *PIK3CA* E545 missense mutations or *PTEN* loss, have been connected with resistance to anti-HER2 treatment[5–9]. To identify additional resistance mechanisms and to avoid the above confounding factors, we selected the HER2+ breast cancer cell lines BT474, SKBR3, and UACC812, which are known to possess neither of those alterations. Examination of *PP2R2B* and other close members of PP2A subunits showed that *PPP2R2B* expression levels exhibited a vast difference in these cell lines, being

highest in BT474 cells but at markedly lower in SKBR3 and UACC812 cells (Fig. 2a). In contrast, the expression of the other examined PP2A subunits showed no significant difference between these cell lines.

We next assessed the response of the three HER2+ cell lines to anti-HER2 agents, including trastuzumab (HER2 targeting antibody, also referred to as Herceptin) and lapatinib, a HER2 kinase inhibitor. The results showed that BT474, which expresses a higher level of *PPP2R2B*, exhibited robust growth inhibition following

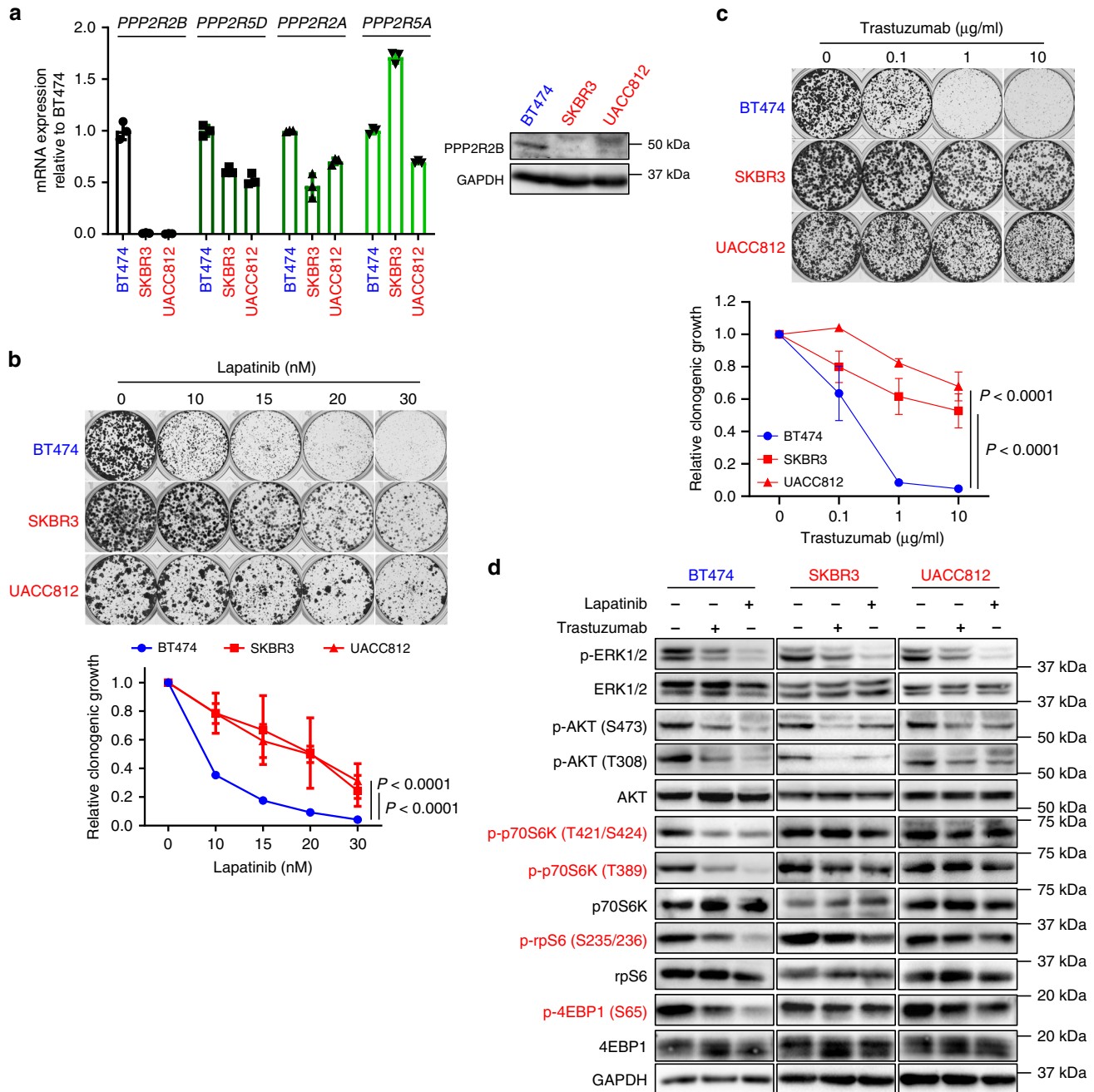

**Fig. 2 Downregulation of *PPP2R2B* in intrinsic resistance to HER2-targeted therapies in cellular models. a** Representative RT-qPCR (left; $n = 2$) and western blot (right; $n = 2$) assessing the expression of the indicated PP2A members in the indicated cell lines. RT-qPCR data are expressed as mean ± s.d. of technical triplicates. **b** Representative images (top) and quantification (bottom) of clonogenic assay for the responses of the indicated cell lines to lapatinib. **c** Representative images (top) and quantification (bottom) of clonogenic assay for the responses of the indicated cell lines to trastuzumab. Data in **b** and **c** are expressed as means ± s.d. of three independent experiments. $P$ values in **b** and **c** were determined by two-way ANOVA. **d** Representative western blot analysis ($n = 2$) of signaling pathways downstream of HER2 with the indicated cell lines treated with or without 10 µg/ml trastuzumab or 40 nM lapatinib. Effectors downstream of mTOR were highlighted in red. The cell line sensitive to HER2 inhibition (BT474) is labeled in blue, whereas the ones resistant (SKBR3 and UACC812) are in red.

treatment with either trastuzumab or lapatinib (Fig. 2b, c). In contrast, SKBR3 and UACC812, which express lower levels of *PPP2R2B*, were relatively resistant to the treatments (Fig. 2b, c). This observation in cellular models is consistent with the hypothesis that *PPP2R2B* downregulation correlates with resistance to anti-HER2 therapies.

The PI3K/AKT/mTOR signaling pathway has previously been implicated in anti-HER2 response[5–9], and several key effectors of the pathway, including AKT, p70S6K, and 4EBP1, are known

targets of PP2A for dephosphorylation[25–30]. Although trastuzumab or lapatinib effectively blocked the phosphorylation of AKT and ERK1/2 in all the three cell lines (Fig. 2d), their inhibition of phosphorylation of p70S6K, ribosomal protein S6 (rpS6), and 4EBP1 was only obvious in sensitive BT474 cells, not in resistant SKBR3 and UACC812 cells (Fig. 2d). These findings established a correlation of *PPP2R2B* with mTOR targets p70S6K and 4EBP, supporting the notion that *PPP2R2B* downregulation leads to impaired ability of trastuzumab or lapatinib to inhibit mTOR

signaling, resulting in refractoriness to anti-HER2 treatments. In contrast, AKT or MAPK activity did not seem to correlate with anti-HER2 resistance in our study.

**Direct evidence of *PPP2R2B* modulating mTOR activity and response to anti-HER2 treatment**. To directly validate the function of *PPP2R2B*, we forced the expression of *PPP2R2B* in SKBR3 cells that express low levels of *PPP2R2B*. Ectopic PPP2R2B promoted growth inhibition and resensitized the anti-HER2-resistant SKBR3 cells to lapatinib (Fig. 3a, b) and trastuzumab (Supplementary Fig. 2a). Consistent with the phenotype, western blot analysis showed that despite modest lapatinib or trastuzumab inhibition of mTOR activity (p70S6K, rpS6, 4EBP1) in the control SKBR3 cells, the drugs more effectively abolished mTOR signals in SKBR3 cells overexpressing *PPP2R2B* (Fig. 3c and Supplementary Fig. 2b). A FLAG-tagged PPP2R2B was co-immunoprecipitated with the scaffolding A subunit and catalytic C subunit (Fig. 3d), indicating the integrity of PPP2R2B-associated PP2A complex. To validate that the p-p70S6K and p-4EBP1 are the substrates of PPP2R2B-PP2A complex, we performed in vitro phosphatase assay in which the immunoprecipitated PPP2R2B-PP2A complex was incubated with the SKBR3 protein extracts to enable an in vitro dephosphorylation event. As expected, the p-p70S6K and p-4EBP1 were markedly reduced by the PPP2R2B-associated PP2A immunoprecipitates (Fig. 3d). In contrast, p-AKT, p-p44/42 MAPK, and p-Myc were not affected by the same immunoprecipitates in the assay (Fig. 3d). This result further supports that PPP2R2B-associated PP2A has a specific activity towards p70S6K and 4EBP1 substrates in HER2+ breast cancer cells.

Conversely, knockdown of *PPP2R2B* in sensitive BT474 cells resulted in resistance to both lapatinib and trastuzumab (Fig. 3e and Supplementary Fig. 2d), increased the levels of p-p70S6K, p-rpS6, and p-4EBP1 and, more obviously, rescued lapatinib-induced inhibition on phosphorylation of p70S6K, rpS6, and 4EBP1 (Fig. 3f). Importantly, the knockdown of *PPP2R2B* did not noticeably change the expression levels of other PP2A regulatory subunits (Supplementary Fig. 2c), suggesting that *PPP2R2B* knockdown does not affect the other PP2A complexes. Together, these gain-of-function or loss-of-function studies provide direct evidence that *PPP2R2B* is a modulator of mTOR activity and a crucial determinant of anti-HER2 response.

**EZH2 and associated chromatin repression mediate *PPP2R2B* downregulation**. We previously reported *PPP2R2B* silencing by DNA hypermethylation in colorectal cancer[24]. To investigate the epigenetic mechanism leading to *PPP2R2B* downregulation in breast cancer, we treated SKBR3 and UACC812 cells with a series of epigenetic compounds including DNA methyltransferase (DNMT) inhibitor 5-aza-2′-deoxycytidine (5AZA), HDAC inhibitor Trichostatin A (TSA), and histone methyltransferase EZH2 inhibitors EPZ-6438 (hereafter EPZ) and GSK126. Among these epigenetic inhibitors, EPZ and GSK126, but not the other two compounds, markedly induced the expression of *PPP2R2B* in both SKBR3 and UACC812 cells (Fig. 4a, b), though they did not affect *PPP2R2A* (Fig. 4a). In addition, pyrosequencing analysis of the *PPP2R2B* promoter did not show evidence of DNA hypermethylation in SKBR3 cells compared with BT474 cells (Supplementary Fig. 3). This finding suggests a role for EZH2, rather than DNA hypermethylation, in suppressing *PPP2R2B* expression in breast cancer.

Further evidence of EZH2-mediated *PPP2R2B* repression was provided by a chromatin immunoprecipitation (ChIP) assay, which showed significantly higher enrichment of EZH2 and associated histone marker H3K27me3 at the *PPP2R2B* promoter region in SKBR3 cells compared to BT474 cells (Fig. 4c), and EPZ treatment abolished the enrichment of H3K27me3 (Fig. 4d). Moreover, *EZH2* knockdown or overexpression of a SET domain-deleted EZH2, which functions as a dominant-negative inhibitor of EZH2[40], also induced the expression of *PPP2R2B*, but not *PPP2R2A* (Fig. 4e and Supplementary Fig. 4). *EZH2* knockdown also enabled lapatinib to effectively inhibit the phosphorylation of p70S6K, rpS6, and 4EBP1 (Fig. 4f) and enhance the growth inhibition of lapatinib (Fig. 4g). Collectively, these findings suggest that EZH2 directly represses *PPP2R2B* expression through chromatin modification, and small molecule inhibition of EZH2 restored *PPP2R2B* expression and abolished the residual p70S6K and 4EBP1 activity in breast cancer.

**Combination of EZH2 inhibitor and anti-HER2 therapy elicits a robust anti-cancer effect in vitro and in vivo**. The above results suggest that EZH2 might be a therapeutic target in the context of anti-HER2 resistance and that EZH2 inhibitor, which acts to restore *PPP2R2B* expression, could overcome resistance to anti-HER2 treatments. To this end, we tested the combined effect of EPZ and anti-HER2 treatments in anti-HER2 resistant cell lines. Co-treatment of EPZ and trastuzumab or lapatinib showed a robust combinatorial effect in abrogating the colony growth of SKBR3 and UACC812 cells, which are otherwise resistant to lapatinib or trastuzumab (Fig. 5a, b). The combination also induced strong apoptosis, as evidenced by subG1 analysis (Supplementary Fig. 5). As expected, the drug combination resulted in a marked reduction of p70S6K, 4EBP1, and rpS6 phosphorylation compared to lapatinib or trastuzumab treatment alone (Fig. 5c, d). Importantly, we demonstrated that the combined effect was mediated through EPZ-induced *PPP2R2B* expression, as *PPP2R2B* knockdown markedly abolished this combinatorial effect (Fig. 5e). The combination effect was also seen using another EZH2 inhibitor GSK126 (Supplementary Fig. 6).

To evaluate the combination of EPZ and trastuzumab in vivo, we used nude mice engrafted with UACC812 cells, which are trastuzumab-resistant. The xenograft tumors show limited response to EPZ or trastuzumab single-agent treatment, while the combination treatment abrogated tumor growth with minimum toxicity to the animal (Fig. 6a and Supplementary Fig. 7, left). Of significant notice, the majority of the tumors treated with EPZ or trastuzumab remained progressive. In contrast, a substantial number of tumors treated with the combination showed either partial response (PD) or complete regression (CR) (Fig. 6a). Immunohistochemistry (IHC) analysis confirmed that the residual tumors, upon the combination treatment, exhibited marked ablations of the p70S6K and 4EBP1 phosphorylation compared to untreated or single-agent treated tumors (Fig. 6c).

As the SKBR3 cells could not form proliferative xenografts in our animal model, we utilized another HER2+ breast cancer cell line MB361 to validate the observation with the UACC812 xenografts. We found that MB361 cells also possess a lower level of *PPP2R2B* compared to BT474 (Supplementary Fig. 8a), and are resistant to trastuzumab (Supplementary Fig. 8c). Importantly, EPZ reactived expression of *PPP2R2B* (Supplementary Fig. 8b) also in MB361 cells and mitigated their resistance to the anti-HER2 antibody (Supplementary Fig. 8c). As expected, the MB361 xenograft model also showed a striking combinational effect from EPZ and trastuzumab when compared to the untreated or single-agent-treated (Fig. 6b), and no bodyweight loss was observed with the combinational treatment in the animal model (Supplementary Fig. 7, right).

Together, the data obtained in the mouse models suggest that the anti-HER2 therapy combined with EZH2 inhibitor can deliver improved efficacy.

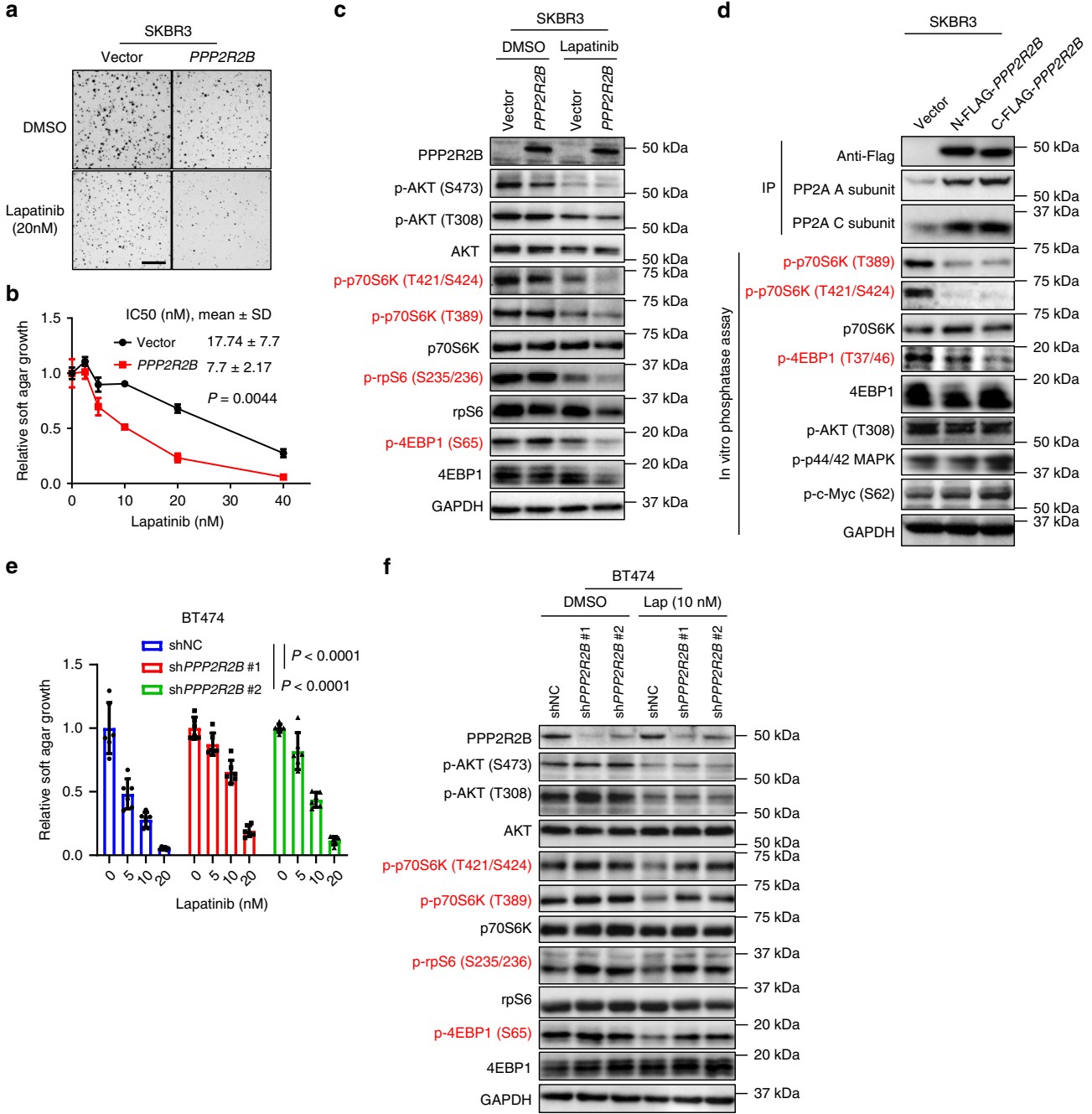

**Fig. 3 Functional validation of *PPP2R2B* in modulating response to HER2-targeted therapies. a** Representative images (*n* = 9) of soft agar assay with SKBR3 cells transduced with either the empty vector (Vector) or the vector containing *PPP2R2B*, and treated with the indicated concentration of lapatinib. Scale bar: 1.5 mm. **b** Quantification of soft agar assay performed with cells from **a** and treated with the indicated concentrations of lapatinib. Data are expressed as means ± s.d. of technical triplicates and representative of three independent overexpression experiments. The indicated IC50s were calculated with the three independent experiments and the *P* value was determined by comparing the IC50s from Vector to the ones from *PPP2R2B*, with two-tailed Student's *t*-test. **c** Representative western blot analysis (*n* = 2) with cells from **a** treated with lapatinib at 40 nM. **d** PPP2R2B-PP2A in vitro phosphatase assay (*n* = 1) using anti-FLAG immunoprecipitates (IP) from SKBR3 expressing empty vector (Vector), N-FLAG-*PPP2R2B*, or the C-FLAG-*PPP2R2B*, incubated with whole cell lysate from SKBR3. **e** Soft agar assay with BT474 transduced with shNC or shRNAs against *PPP2R2B* and treated with the indicated concentrations of lapatinib. Data are expressed as mean ± s.d. of two independent experiments performed in triplicate (*n* = 6). *P* values were determined by two-way ANOVA. **f** Representative western blot analysis (*n* = 2) with cells from **d** treated with lapatinib (Lap) at 10 nM. Effectors downstream of mTOR were highlighted in red in **a**, **d**, and **e**.

***PPP2R2B* downregulation is associated with acquired resistance of anti-HER2 therapy.** In the clinic, metastatic tumors initially sensitive to HER2-targeted therapies almost inevitably acquire resistance[3,4]. Therefore, we investigated whether epigenetic repression of *PPP2R2B* by EZH2 also contributes to the acquisition of anti-HER2 resistance. To establish acquired resistance, we treated the BT474 cells with increasing concentrations of lapatinib or trastuzumab for 3–5 months until they maintained their viability even in the presence of high concentrations of the drugs (100 nM lapatinib or 10 μg/ml trastuzumab). The derived

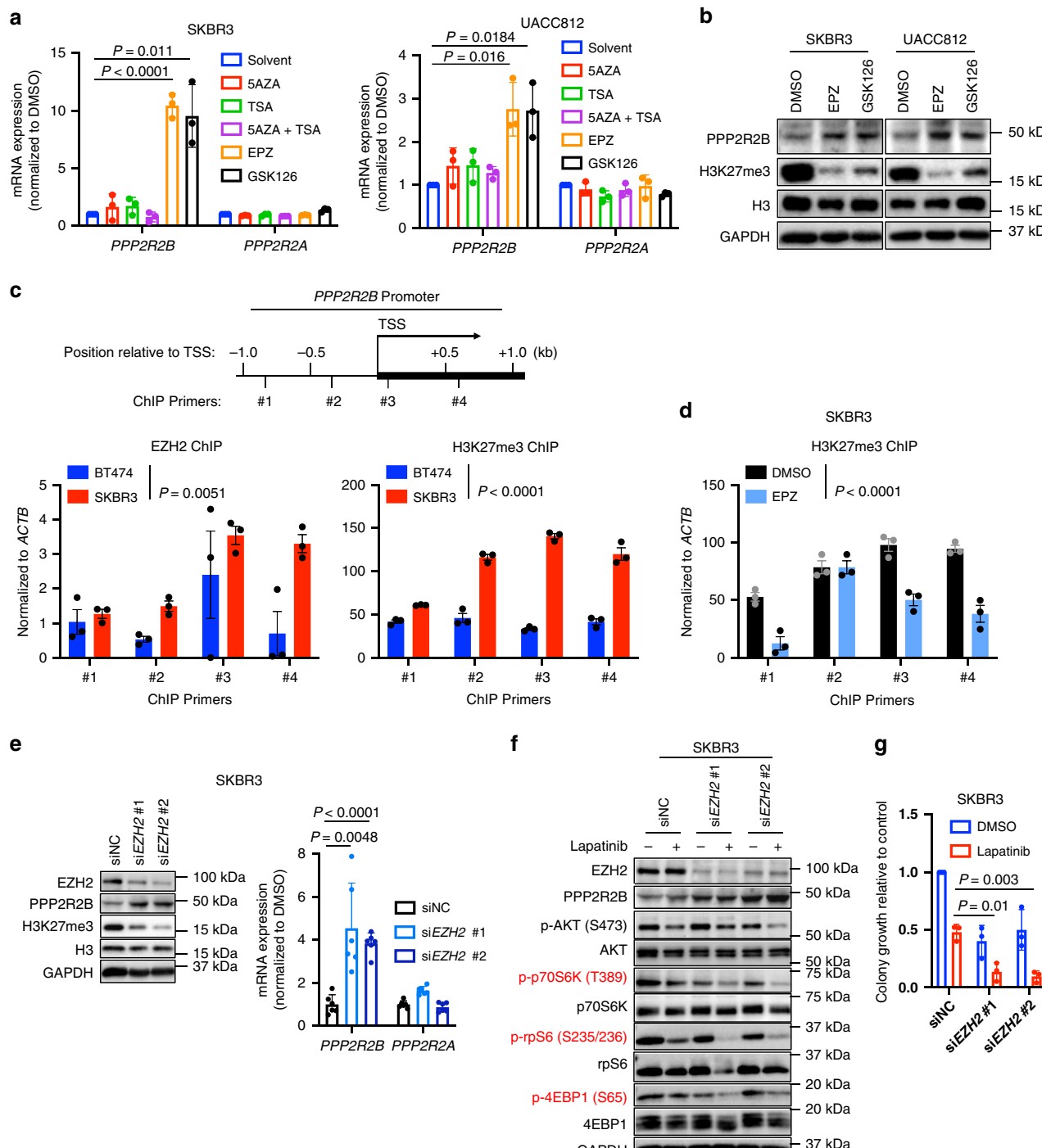

**Fig. 4 PPP2R2B is repressed by EZH2 and EZH2 inhibitor treatment restores PPP2R2B expression. a** RT-qPCR analysis of *PPP2R2B* and *PPP2R2A* in the indicated cell lines treated with the indicated epigenetic compounds. RT-qPCR data are expressed as mean ± s.d. of technical triplicates. **b** Western blot analysis ($n = 1$) of the indicated proteins in cells treated with EPZ or GSK126. **c** Top panel: schematic showing the ChIP primer locations with respect to the transcriptional start site (TSS) of *PPP2R2B* promoter. Bottom panel: ChIP-qPCR of EZH2 (left) and H3K27me3 (right) enrichment at *PPP2R2B* promoter in BT474 and SKBR3, using the four pairs of primers shown at the top. **d** ChIP-qPCR of H3K27me3 enrichment on *PPP2R2B* promoter in SKBR3 treated with DMSO or 1 μM EPZ for seven days, with the primers shown in **c**. Data in **c** and **d** are expressed as mean ± s.e.m. of technical triplicates. *P* values in **c** and **d** were determined by two-way ANOVA. **e** Western blot analysis (left) and RT-qPCR (right) in SKBR3 transfected with siNC or siRNAs against *EZH2*. qPCR data are expressed as mean ± s.e.m. of two independent experiments performed in triplicate ($n = 6$). **f** Western blot analysis with cells from **e** treated with 40 nM lapatinib. Effectors downstream of mTOR were highlighted in red. **g** Soft agar assay with cells from **e** treated with 20 nM lapatinib. Data are expressed as mean ± s.d. of three independent knockdown experiments. *P* values in **a**, **e**, and **g** were determined with two-tailed Student's *t*-test, and corrected with Bonferroni adjustment.

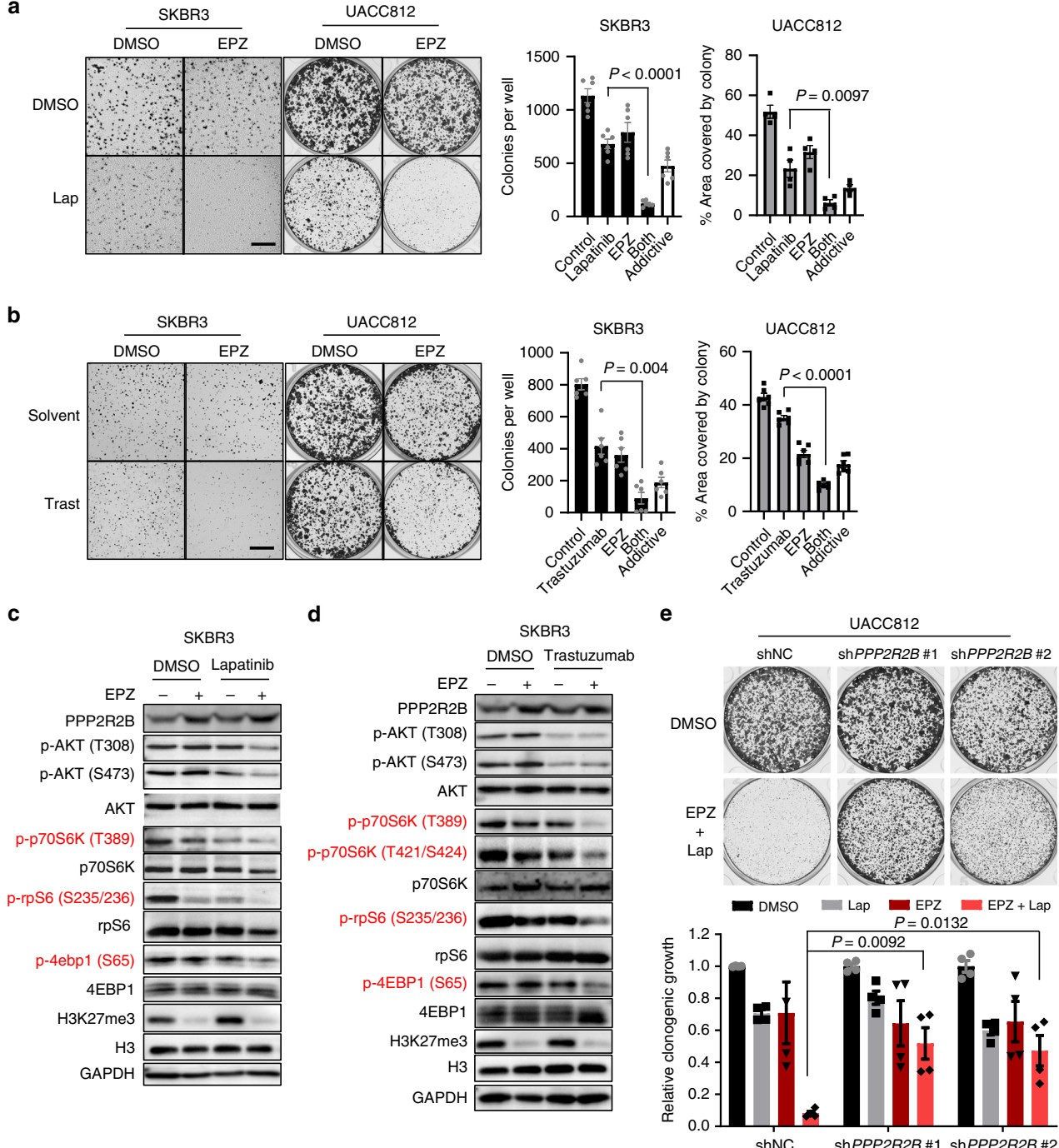

**Fig. 5 EPZ and anti-HER2 combination elicits robust effect in vitro. a** Representative images (left) and quantification (right) of soft agar assay with SKBR3 or clonogenic assay with UACC812 treated with DMSO, EPZ (1 μM), lapatinib (lap; 20 nM), or the combination of both. Data are expressed as mean ± s.e.m. of two independent experiments performed in triplicate for SKBR3 ($n = 6$) and duplicate for UACC812 ($n = 4$). Scale bar: 1.5 mm. The addictive effect from the EPZ and lapatinb was computed and shown in the figure. **b** Representative images (left) and quantification (right) of soft agar assay with SKBR3, or clonogenic assay with UACC812 treated with the solvent, EPZ (1 μM), trastuzumab (trast; 10 μg/ml), or the combination of both. Data are expressed as mean ± s.e.m. of two independent experiments performed in triplicate ($n = 6$). Scale bar: 1.5 mm. The addictive effect from the EPZ and trastuzumab was computed and shown in the figure. *P* values in **a** and **b** were determined with two-tailed Student's *t*-test. **c** Representative western blot analysis ($n = 2$) with SKBR3 treated with the solvent, lapatinib (40 nM), EPZ (1 μM), or the combination of both. **d** Representative western blot analysis ($n = 2$) with SKBR3 treated with the solvent, trastuzuamb (10 μg/ml), EPZ (1 μM), or the combination of both. Effectors downstream of mTOR were highlighted in red in **c** and **d**. **e** Representative images (top) and quantification (bottom) of clonogenic assay with UACC812 transduced with shNC or shRNAs against *PPP2R2B* and treated with DMSO, lapatinib (lap; 20 nM), EPZ (1 μM), or the combination of both. Data are expressed as mean ± s.e.m. of two independent experiments performed in duplicates ($n = 4$). *P* values were determined with two-tailed Student's *t*-test, and corrected with Bonferroni adjustment.

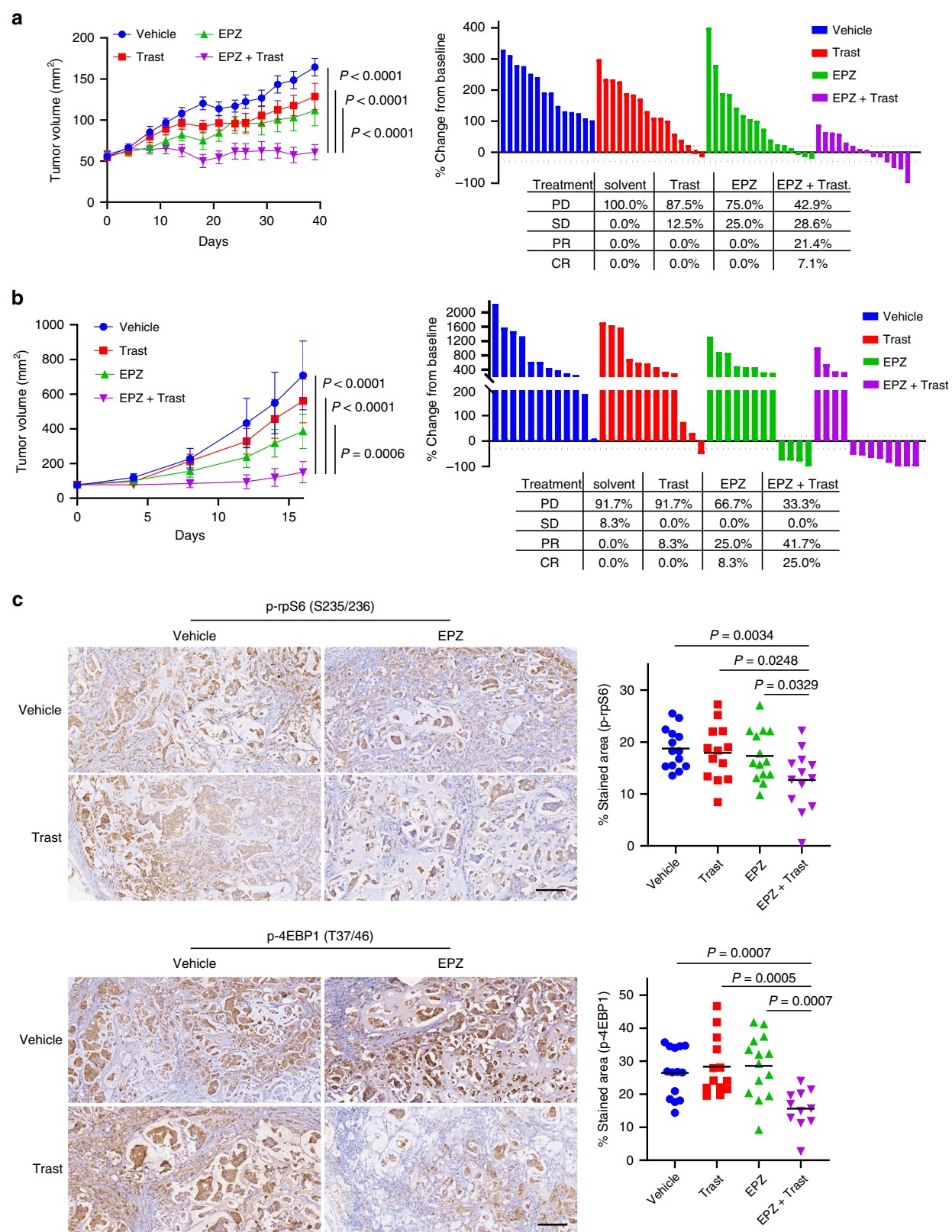

lapatinib-resistant cell line is denoted as BT474LR (Fig. 7a), and the trastuzumab-resistant cell line is denoted as BT474TR (Fig. 7b). As expected, the expression of *PPP2R2B*, but not the other PP2A subunits, showed marked downregulation in both BT474LR and BT474TR cells compared to parental BT474 cells (Fig. 7c). Accordingly, forced expression of *PPP2R2B* in the acquired resistance cells re-sensitized them to lapatinib treatment (Supplementary Fig. 9a). Consistent with this, BT474LR cells showed enhanced levels of p70S6K, rpS6 and 4EBP1 phosphorylation compared to BT474 cells (Supplementary Fig. 9b), leading to refractoriness to lapatinib treatment that was, however, overcome by *PPP2R2B* overexpression (Supplementary Fig. 9b).

**Fig. 6 Combinational effect of trastuzumab and EPZ in vivo. a** Tumor volume along treatment (left) or change (%; right) of tumor volume from baseline tumors in NCR nude mice bearing UACC812 xenografts were treated with vehicle ($n = 7$), EPZ ($n = 8$), trastuzumab (trast; $n = 8$), or in combination (EPZ + Trast; $n = 7$). EPZ was administered once per day via oral gavage at 250 mg/kg, while trastuzumab was administered once per week via intraperitoneal injection at a loading dose of 30 mg/kg and subsequently 15 mg/kg. Tumor volumes were measured twice a week. **b** Tumor volume along treatment (left) or change (%; right) of tumor volume from baseline tumors in NCR nude mice bearing MB361 xenografts were treated with vehicle ($n = 7$), EPZ ($n = 6$), trastuzumab (trast; $n = 6$), or in combination (EPZ + Trast; $n = 6$). EPZ was administered once per day via oral gavage at 250 mg/kg, while trastuzumab was administered once every six days via intraperitoneal injection at a loading dose of 30 mg/kg and subsequently 15 mg/kg. *P* values in **a** and **b** were determined by two-way ANOVA and adjusted with Bonferroni correction. **c** Representative images (left) and quantification (right) of IHC with tumors from **a**. Data were acquired with tumors from seven mice ($n = 7$) from each group. Scale bar: 150 μm. *P* values were determined with two-tailed Student's *t*-test.

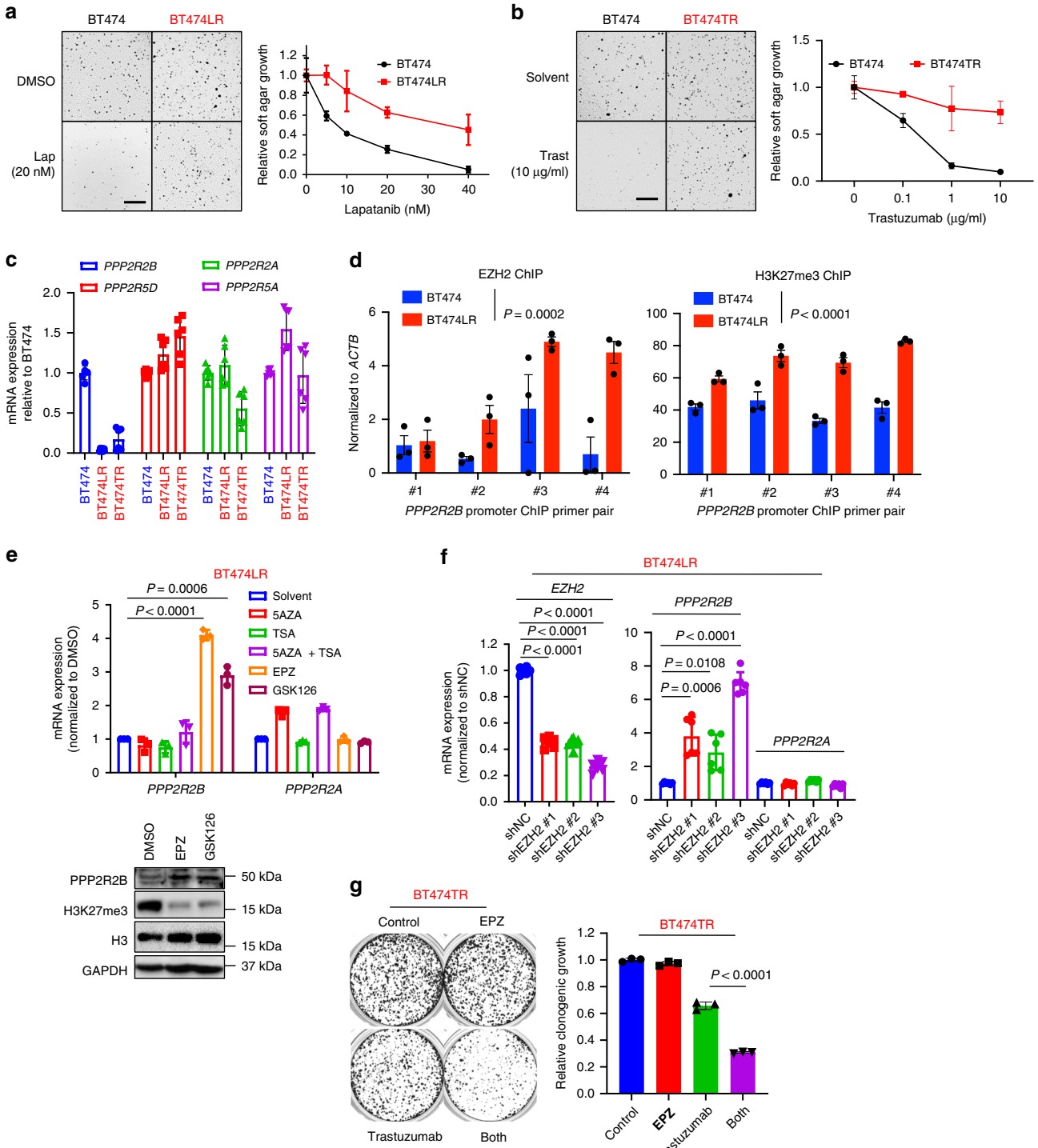

**Fig. 7 Downregulation of *PPP2R2B* in acquired resistance to HER2-targeted therapies. a** Representative images (left) and quantification (right) of soft agar assay with BT474 and BT474LR (lapatinib resistant) treated with increasing concentrations of lapatinib (lap). **b** Representative images (left) and quantification (right) of soft agar assay with BT474 and BT474TR (trastuzumab resistant) treated with increasing concentrations of trastuzumab (trast). Data in **a** and **b** are expressed as mean ± s.d. of technical triplicates, and representative of three independent experiments. **c** Expression of the indicated PP2A members assessed by RT-qPCR in BT474, BT474LR, and BT474TR. Values represent mean ± s.d. of two independent experiments performed in triplicate ($n = 6$). **d** ChIP-qPCR of EZH2 (left) and H3K27me3 (right) enrichment on *PPP2R2B* promoter in BT474 and BT474LR, using the four pairs of ChIP primers flanking the promoter. Data are expressed as mean ± s.e.m. of technical triplicates. $P$ values were calculated with two-way ANOVA. **e** RT-qPCR (top) and western blot analysis (bottom) with BT474LR treated with the indicated compounds. Values represent mean ± s.d. of technical triplicates. **f** RT-qPCR assessing expressions of the indicated genes with BT474LR carrying shNC or shRNAs against *EZH2*. Values represent mean ± s.d. of two independent experiments performed in triplicate ($n = 6$). $P$ values in **e** and **f** were determined with two-tailed Student's $t$-test, and corrected with Bonferroni adjustment. **g** Representative images (left) and quantification (right) of clonogenic assay with BT474TR treated with the solvent, EPZ (2 μM), trastuzumab (10 μg/ml), or in combination at the indicated concentrations. Values represent mean ± s.d. of technical triplicates. Data are representative of two independent experiments. $P$ value was determined with two-tailed Student's $t$-test. The parental cell line is labeled in black, whereas the resistant ones are in red.

Furthermore, the ChIP assay showed increased enrichment of EZH2 and H3K27me3 in *PPP2R2B* in BT474LR compared to BT474 cells (Fig. 7d). Accordingly, the EZH2 inhibitor EPZ or GSK126 (but not other epigenetic inhibitors), or *EZH2* knockdown induced the expression of *PPP2R2B* in acquired resistant cells (Fig. 7e, f), and strongly inhibited their growth when combined with trastuzumab (Fig. 7g). Collectively, these findings demonstrated the involvement of EZH2-mediated *PPP2R2B* suppression in acquired resistance to anti-HER2 treatments.

**Clonal cellular heterogeneity accounts for acquired resistance to anti-HER2 treatments.** Tumor heterogeneity contributes to the acquisition of drug resistance. Having shown a role for EZH2-mediated *PPP2R2B* repression in acquired resistance to anti-HER2 treatments, we next sought to examine whether the pre-existing epigenetic heterogeneity in subpopulations of initially sensitive bulk cells conferred the development of resistance and whether the existence of such a mechanism would provide new insights into treatment strategies. We examined whether the BT474 cells, though sensitive at the bulk level, carried heterogeneous subclones that expressed different levels of *PPP2R2B* and thus conferred differential responses to anti-HER2 treatment. We isolated single-cells from BT474 through single-cell picking under a microscope and expanded these to a panel of ten lines of single cell-derived clones. Among these clones, clones #3 and #4 showed drastic downregulation of *PPP2R2B* expression compared to other clones, though the other candidate PP2A subunits only showed modest variations across the different clones (Fig. 8a). To test whether clones #3 and #4 represented drug-tolerant subclones that could survive otherwise lethal drug exposure, we subjected them to 100 nM lapatinib for three weeks. Indeed clones #3 and #4 showed substantially higher drug tolerance than other clones (Fig. 8b). Knockdown of *PPP2R2B* in BT474 cells with high *PPP2R2B* expression induced a drug-tolerant phenotype against high concentrations of trastuzumab or lapatinib (Fig. 8c). Furthermore, the ChIP assay showed that clones #3 and #4 had higher enrichments of EZH2 and H3K27m3 at *PPP2R2B* compared with the two other clones with higher expression of *PPP2R2B* (Fig. 8d), and consistently EPZ treatment resulted in induction of *PPP2R2B* in clones #3 and #4 (Fig. 8e). Cumulative data from the single-cell clones suggested the existence of a subpopulation of drug-tolerant cells characterized by low expression of *PPP2R2B* and enhanced chromatin enrichment of EZH2 and H3K27me3, contributing to the ultimate emergency of anti-HER2 resistance.

Given the above observations, we propose that initial co-treatment with EPZ might inhibit the acquisition of resistance to trastuzumab by targeting the pre-existing low-*PPP2R2B* expressing clones. To achieve this, we treated BT474 cells with trastuzumab for 25 days to nearly completely eliminate cell growth, which nevertheless robustly recovered after 20 days of drug wash-off. However, an initial EPZ co-treatment with trastuzumab largely abolished cell recovery (Fig. 8f). These data suggest that an upfront co-treatment strategy with an EZH2 inhibitor could prevent the emergence of resistance to HER2-targeted therapy, which is believed to impair the viability of residual drug-tolerant subclones, thus diminishing the recurrence of therapy-refractory cancer cells.

## Discussion

Epigenetic regulation enables tumors to respond to changing environments during progression and facilitates treatment resistance. In this study, we discovered an epigenetic mechanism leading to resistance to anti-HER2 therapy that is mediated by EZH2-mediated PP2A inhibition. We demonstrated that PP2A regulatory subunit *PPP2R2B* is a major subunit of PP2A whose downregulation is associated with poor prognosis in HER2+ breast cancer and resistance to anti-HER2 treatments, including both HER2-targeting antibody trastuzumab and HER2 kinase inhibitor lapatinib. Importantly, we showed that a clinically-available EZH2 inhibitor could restore the expression of *PPP2R2B* to allow effective treatment of HER2-targeted therapies. We thus conclude that epigenetic repression of *PPP2R2B* by EZH2 is a potential mechanism of resistance to anti-HER2 therapy and that EZH2 inhibitors have potential utility to overcome anti-HER2 resistance (Fig. 9).

This work revealed a crucial role of PP2A in HER2+ breast cancer, particularly in the context of anti-HER2 therapy. PP2A subunits and isoforms are encoded by numerous genes, assembling different PP2A complexes that dephosphorylate different target proteins and so function distinctly[32] to exert a role on tumor-suppressors[23] or modulating drug sensitivity[37]. The observation that *PPP2R2B*, among all the PP2A subunits, is most prognostic in HER2+ breast cancer, supports its role in impacting the treatment outcome of trastuzumab. Interestingly, *PPP2R2B* silencing by DNA hypermethylation has previously been found to be associated with resistance to mTOR inhibitor treatment[24]. Therefore, the epigenetic mechanism of *PPP2R2B* silencing appears to be cancer type-dependent, highlighting the need for different epigenetic compounds to activate *PPP2R2B* expression in different cancer types. Interestinglhy, a recent report shows that EZH2 inhbitor can activatate type-I interferon signaling to enable immune response and thus improve a mouse anti-Erbb2 efficacy in a immunocompetent mouse model[41]. Our combination effect however, is different from this mechanism as it occurs in immune-decifcent nude mice. Therefore, the comination of EZH2 inhibitor with

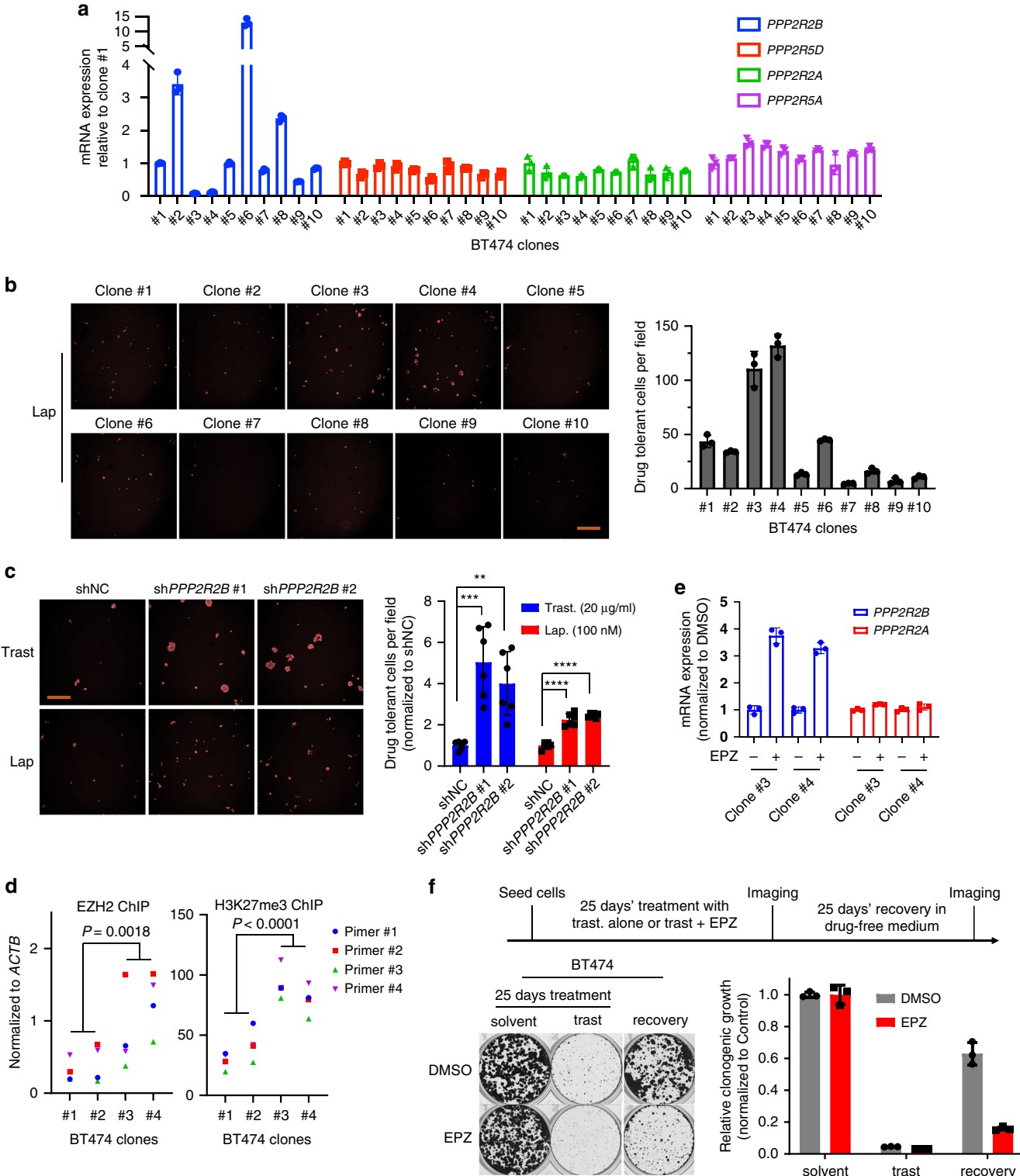

anti-HER2 can elicit the anti-tumor effect through modulating both the tumor instrinic and immune microenvironment mechanisms.

Currently, only a handful of clinical studies have investigated the efficacy of a combination strategy to overcome resistance to anti-HER2 treatment, with limited success. Although the implication of PI3K/AKT/mTOR signaling in resistance to anti-HER2 therapies is promising, clinical trials exploring the combination of trastuzumab with inhibitors of this pathway such as the

mTOR inhibitor everolimus has not delivered the expected improvements[14,15]. Our study shows that anti-HER2 resistance is associated with residual phosphorylation of both p70S6K and 4EBP1, but not with other known PP2A substrates such as AKT, MYC, or ERK. PP2A is known to interact with, and directly dephosphorylate, mTOR targets p70S6K and 4EBP1[26–28]. We confirmed that PPP2R2B-associated PP2A targets p70S6K and 4EBP1 for dephosphorylation, and thus reduced expression of PPP2R2B in breast cancer cells is associated with diminished

**Fig. 8 Clonal heterogeneity contributes to acquired resistance to anti-HER2 treatments. a** Representative RT-qPCR ($n = 2$) assessing expression of the indicated PP2A members in the indicated single cell–derived clones from BT474. Data are expressed as mean ± s.d. of technical triplicates. **b** Left panel: representative images ($n = 6$) where the cells are pseudo-colored red from drug-tolerant cell assay with the ten single cell–derived clones from BT474 treated with lapatinib (lap; 100 nM) for three weeks. Scale bar: 150 μm. Right panel: quantification of the assay on the top. Data are expressed as mean ± s. d. of technical triplicates and representative of two independent experiments. **c** Representative images (left; $n = 6$) where the cells are pseudo-colored red and quantification (right) of drug-tolerant cell assay with BT474 carrying shNC or shRNAs against *PPP2R2B* and treated with trastuzumab (trast; 20 μg/ml) or lapatinib (lap; 100 nM) for three weeks. Data are expressed as mean ± s.d. of two independent experiments performed in triplicate ($n = 6$). Scale bar: 150 μm. *P* values were determined with two-tailed Student's *t*-test, and corrected with Bonferroni adjustment. \*\**P* = 0.0032 (trast-treated shNC vs. sh*PPP2R2B* #2), \*\*\**P* = 0.0008 (trast-treated shNC vs. sh*PPP2R2B* #1), \*\*\**P* < 0.0001 (lap-treated shNC vs. sh*PPP2R2B* #1, or vs. sh*PPP2R2B* #2). **d** ChIP-qPCR of EZH2 (left) and H3K27me3 (right) enrichments on *PPP2R2B* promoter in the indicated single cell–derived clones from BT474, using the four pairs of ChIP primers flanking the promoter. Data are expressed as mean values of technical triplicates. *P* values were determined using two-way ANOVA. **e** RT-qPCR assessing expression of *PPP2R2B* and *PPP2R2A* in single cell–derived clone #3 and #4, treated with or without EPZ. Data are expressed as mean ± s.d. of technical triplicates. **f** Top panel: schematic showing how the cell regrowth assay was performed. Briefly, the cells were treated with trastuzumab (trast.) at 1 μg/ml alone or the combination of EPZ (2 μM) and trast. for 25 days, followed by 20 days' recovery in drug-free medium. Imaging and quantification were performed after the treatment and recovery. Bottom panel: representative images (left) and quantification (right) of the cell regrowth assay shown in the top panel with BT474. Data are expressed as mean ± s.d. of technical triplicates and representative of two independent experiments.

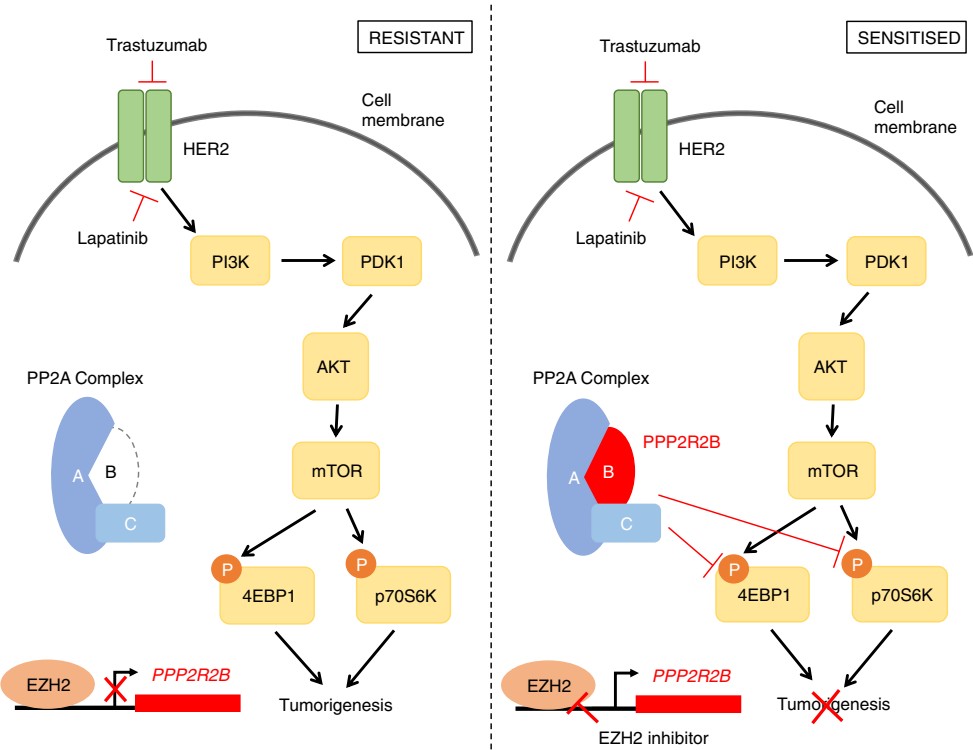

**Fig. 9 Proposed model for the role of *PPP2R2B* in determining the sensitivity to HER2-targeted therapies.** EZH2-mediated repression (left) of *PPP2R2B* leads to the absence of functional PPP2R2B-containing PP2A, and thus residual p-4EBP1 and p-p70S6K and tumorigenesis, with HER2-targeted therapies. Accordingly, inhibition of EZH2 (right) restores *PPP2R2B* expression and functional PPP2R2B-containing PP2A that reduces phosphorylation of 4EBP1 and p70S6K and thus inhibits tumorigenesis under anti-HER2 treatments.

p70S6K and 4EBP1 dephosphorylation by anti-HER2 treatment. Mechanistically, p70S6K and 4EBP1 regulate two parallel downstream pathways of mTOR[42,43] that are crucial for cancer cell proliferation and tumorigenesis[42,43], as simultaneously inhibition of both pathways prevents cancer cell cycle progression[43]. We suggest that persistent residual signaling of both p70S6K and 4EBP1 following anti-HER2 treatment in low *PPP2R2B*-expressing cancer cells might collaterally confer treatment failure, though its correlation with clinical resistance to anti-HER2 therapy needs futher validated in clinical samples. Therefore, enhancing PP2A activity by restoring *PPP2R2B* expression to enable a complete inhibition of both p70S6K and 4EBP1 phosphorylation appears to be an efficient means to augment anti-HER2 treatment.

Our study demonstrates that EZH2-mediated *PPP2R2B* silencing is required for both drug tolerance and acquired resistance to anti-HER2 treatments, as supported by clinical analysis showing a potential predictive value of *PPP2R2B* in HER2-targeted therapy. Single clone analysis of BT474 cells that are primarily sensitive to anti-HER2 treatment revealed heterogeneity of *PPP2R2B* expression, with low *PPP2R2B*-expressing subclones showing higher enrichment of EZH2 and H3K27me3 in the *PPP2R2B* promoter and higher tolerance to anti-HER2 therapies. Therefore, our data support the notion that anti-HER2 resistance may arise from the pre-existing tolerant epigenetic state. Indeed, we show that upfront co-treatment with the EZH2 inhibitor EPZ can prevent the emergency of BT474 resistance following trastuzumab or lapatinib treatments. Therefore, we propose that

inhibiting EZH2-mediated *PPP2R2B* silencing might mitigate anti-HER2 resistance by restoring the vulnerability of the tolerance state. In addition, given the EZH2-mediated cellular heterogeneity towards resistance, as demonstrated in this study, upfront co-treatment with EPZ-6438 and anti-HER2 therapy might prevent tumor recurrence and metastasis. EPZ-6438 is a leading EZH2 inhibitor that is recently in an advanced stage of clinical development. Our findings support future clinical studies to investigate the clinical utility of this combination strategy for treating HER2+ breast cancer while evaluating *PPP2R2B* as a potential predictive marker of anti-HER2 therapy.

## Methods

**Cell lines and reagents**. BT474, SKBR3, MB361, UACC812, and HEK293T were obtained from American Type Culture Collection (ATCC; Manassas, VA) and authenticated according to ATCC instructions. All cell lines were cultured in an incubator set at 37 °C in 5% $CO_2$ and 95% atmospheric air, and maintained with DMEM (Hyclone; cat. no. SH30243.01) supplemented with 10% (v/v) fetal bovine serum (cat. no. Biowest; S181BH-500) and 100 U/ml penicillin-streptomycin (Gibco; cat. no. 15140-122). The acquired-resistant cell line, BT474LR, was generated by culturing the parental line, BT474, in normal growth medium supplemented with 50 nM lapatinib for five months, and subsequently with 100 nM lapatinib for additional four months; while BT474TR in its normal growth medium supplemented with 10 µg/ml trastuzumab for 3 months. All cell lines were assessed regularly to ensure they were free of mycoplasma contamination.

Lapatinib (cat. no. S2111) was purchased from Selleck Chemicals (Houston, Texas) and prepared with DMSO for in-vitro experiments. EPZ-6438 (cat. no. GC14062) was purchased from Glpbio Technology Inc. (Montclair, California) and prepared with DMSO for in-vitro experiments. Trichostatin A (TSA; cat. no. T8552) and 5-aza-2′-deoxycytidine (cat. no. A3656) were purchased from Sigma-Aldrich (St. Louis, Missouri) and prepared with methanol and sterile water, respectively. Trastuzumab and its solvent were acquired from Roche (South San Francisco, California).

**Patient survival analysis**. Kaplan–Meier (KM) survival analysis was performed using KM-Plotter online database for breast cancer microarray (https://kmplot.com/)[44] and Gene expression-based Outcome for Breast cancer Online database (GOBO) (http://co.bmc.lu.se/gobo/)[45]. In KM-Plotter analysis only JetSet best probe sets were used, and the computed based performing thresholds which are used as cut-offs were auto-selected for the percentiles of the subjects between the low and high gene-expression groups. In GOBO analysis, gene symbols were selected as identifiers, as well as the clinical data divided into three quantiles and censoring years was selected as ten years. In both databases, the relationship of gene expression level and relapse-free survival (RFS), distant metastasis-free survival (DMFS) and overall survival (OS) evaluated for each intrinsic subtype of breast cancer (Luminal A, Luminal B, HER2+, and Basal).

**Single-cell clone isolation**. Dissociated BT474 cells were suspended and diluted with growth medium to 2500 cells per ml, and 2 ml of the diluted cell-suspension was plated into a 60 mm tissue culture dish. Each single cell was picked up with a pipette under a microscope and plated into one well of a 96 well dish containing 100 µl growth medium. The cells were then cultured for two to 3 weeks with medium refreshed every five to seven days until they became expandable to dishes with larger well dimensions and eventually T75 flasks.

**Reverse transcription-quantitative PCR (RT-qPCR)**. Cells or tissues were lysed in TRIzol (Invitrogen) and RNA was extracted using the Direct-zol™ RNA Mini-Prep kit (ZymoResearch; cat. no. R2052), according to the user manual. The extracted RNAs were reverse-transcribed into cDNAs using the T100 Thermal Cycler (BIO-RAD) and High-Capacity cDNA Reverse Transcription Kit (Applied Biosystems; cat. no. 4368814), according to the user manual. Quantitative PCR (qPCR) was carried out with the KAPA SYBR FAST qPCR kit (KAPA Biosystems; cat. no. KK4620), according to the user manual. All reactions were performed in technical triplicates using the ViiA7 Real-Time PCR System (Thermo Fisher Scientific) in a 96-well or 386-well plate format. *GAPDH* was used as an internal control. Fold changes of expression were calculated using the $2^{-\Delta\Delta CT}$ method. The gene-specific primers used are listed in Supplement Table 2.

**Preamplification of cDNAs**. Preamplification of cDNA was performed with the TaqMan® PreAmp Master Mix Kit (Thermo Fisher Scientific; cat. no. 4384267) according to the user manual. Briefly, cDNAs equivalent to 4 ng of RNAs were mixed with the pooled qPCR primers and the enzyme-containing Master Mix from the commercial kit. The mixture was then amplified for ten cycles with a thermal cycler. The preamplified cDNAs were then diluted five times with water before subjected to qPCR.

**Western blot**. Cells were lysed with 4% (w/v) sodium dodecyl sulfate (SDS) (1st BASE; cat. no. BUF2051) solution. Protein concentration was determined with the Pierce™ BCA Protein Assay Kit (Thermo Fisher Scientific; cat. no. 23227) according to the user manual. Equal amounts of protein from each sample were separated with SDS-polyacrylamide gel electrophoresis (SDS-PAGE) and transferred to PVDF membranes (Merck Millipore; cat. no. IPVH00010), which were then blocked with appropriate blocking reagent and blotted with the specific primary antibodies according to the manufacturer's instructions. The following primary antibodies, anti-p70S6K (cat. no. 9202), anti-phospho-p70S6K (Thr421/Ser424) (cat. no. 9204), anti-phospho-p70S6K (Thr389) (cat. no. 9234), anti-phospho-AKT (Ser473) (cat. no. 4058), anti-phospho-AKT (Thr308) (cat. no. 2965), anti-AKT (cat. no. 4691), anti-rpS6 (cat. no. 2217), anti-phospho-rpS6 (Ser235/236) (cat. no. 2211), anti-4EBP1 (cat. no. 9644), anti-phospho-4EBP1 (Ser65) (cat. no. 9451), anti-GAPDH (cat. no. 2118), anti-EZH2 (cat. no. 5246), anti-phospho-p44/42 MAPK (cat. no. 9101), anti-histone H3 (cat. no. 9715), anti-PP2A A subunit (cat. no. 2041), and anti-PP2A C subunit (cat. no. 2259) were purchased from Cell Signaling Technology. Anti-PPP2R2B (cat. no. ab16447) and anti-H3K27me3 (cat no. 07-449) antibodies were purchased from Abcam and Merck Millipore, respectively; and anti-phospho-Myc (S62) (cat. no. 33A12E10) was acquired from BioAcademia. For detecting endogenous PPP2R2B, anti-PPP2R2B (LS-C761012) purchased from LSBio was used. After blotting with the primary antibodies, the membranes were incubated with horseradish peroxidase-linked secondary antibody, anti-mouse IgG (NA931-1ML) or anti-Rabbit IgG (NA934-1ML) acquired from GE Healthcare, followed by visializaiton with chemiluminescent substrate (Thermo Fisher Scientific; cat. no. 34096) and the ChemiDocTM XRS + Imaging System (Bio-Rad). All primary antibodies were used in a dilution of 1:1000, expect anti-GAPDH used in 1:2500, and the secondary antibodies were used in 1:4000 dilution. All full blots are provided in the Supplementary file.

**Soft agar assay**. Experiments were carried out with plates coated with a base layer of growth medium containing 0.6% (w/v) agar (Becton Dickinson; cat. no. 214010). Cells were plated at a density of 5000 or 2500 cells per well in growth medium containing 0.3% (w/v) agar on top of the base agar, in 12-well or 24-well plates, respectively. Drugs were administered after plating the cells, with top-up medium. The concentrations of drugs were determined based on the total volume of medium and agar contained in one well. After a total of 14 to 21 days of culturing, colonies were stained with iodonitrotetrazolium chloride (Sigma-Aldrich; cat. no. I10406) at 0.25 mg/ml for 4 h, followed by quantification with GelCount (Oxford Optronix) according to the manufacturer's instructions. IC50 values of the drugs were determined by CompuSyn (downloadable from http://www.combosyn.com/) following the instructions of the developer of the software.

**Clonogenic assay**. Cells were seeded on 6-well or 12-well dishes at optimized densities and treated with the indicated drugs on the next day, for a total of 24–28 days, until the control treated with the solvent reached about 80% confluency. Drugs and medium were refreshed every four to seven days during treatment, and thereafter colonies of cells were fixed with methanol and stained with 0.5% (w/v) crystal violet. Images of the colonies were taken with GelCount (Oxford Optronix) and the percentage of area covered by the colonies in each well was quantified with ImageJ 1.51u (downloaded from https://imagej.net/Downloads). IC50 values of the drugs were determined by CompuSyn 1.0 (downloadable from http://www.combosyn.com/), following the instructions of the developer of the software.

**Sub-G1 apoptosis assay**. Briefly, cells were seeded on six-well dishes and treated with the indicated compounds for two days, after which both the floating and attached cells were harvested. The cells were then fixed with 70% ethanol and washed twice with PBS, followed by treatment with RNase A and staining with propidium iodide. The stained cells were then subjected to flow cytometry to analyze the sub-G1 cell population in comparison to the whole population.

**Drug-tolerant cell assay**. Procedures described by Guler et al.[46] were followed to generate and quantify drug-tolerant cells. Briefly, cells were seeded at optimized densities in 12-well dishes and treated with the indicated drugs on the next day at the indicated concentrations for a total of three weeks. Medium and drugs were refreshed every four to seven days during treatment, after which the remaining cells on the dishes were fixed with methanol and stained with 300 nM 4′,6-diamidino-2-phenylindole (DAPI) (Thermo Fisher Scientific; cat. no. 62248). The stained nuclei were quantified by the imaging system and analysis software from PerkinElmer Harmony 4.8 (Opera Phenix High-Content Screening System) according to the instructions from the manufacturer. For each well, 25 images were acquired at ×5 magnification and the average number of nuclei per image was calculated. Data derived from images that were out of focus were excluded.

**Cell regrowth assay**. Cells were seeded on 12-well dishes at optimized densities and treatment was administered the next day for a total of 25 days. Medium and drugs were refreshed every four to six days during the treatment, after which the medium with drugs were removed and the remaining surviving cells were cultured in drug-free growth medium for a total of 20 days. Colonies of cells were then fixed

with methanol, stained with 0.5% (w/v) crystal violet, and images of the colonies were taken with GelCount 1.2.1.0 (Oxford Optronix). The percentage of areas covered by the colonies on each well was quantified with ImageJ.

**RNA interference**. Cells were transfected with small interfering RNAs (siRNAs) using the Lipofectamine RNAiMAX (Invitrogen) following the manufacturer's instructions, at a final concentration of 100 nM. All siRNAs were purchased from Sigma-Aldrich. The non-targeting control siRNAs (cat. no. SIC001) were designated as siNC, and the siRNAs against EZH2 were designated as siEZH2 and their targeted sequences were listed in Supplement Table 3.

To generate shRNA stable cells, shRNA with the targeted sequence listed in Table 3 was cloned into pLKO.1-puro vector (Addgene; cat. no. 8453) using the protocol available on Addgene (http://www.addgene.org/tools/protocols/plko/). A pair of shRNAs that target firefly luciferase were used as a negative control, designated as shNC. The shRNAs that target PPP2R2B or EZH2 were designated as shPPP2R2B or shEZH2, respectively. To generate virus for transduction, HEK293T cells were transfected with the cloned vector and packaging plasmids pMDLg/pRRE (Addgene; cat. no. 12251), pRSV-REV (Addgene; cat. no. 12253), and pMD2.G (Addgene; cat. no. 12259), with Lipofectamine 2000 (Invitrogen; cat. no. 11668019), according to the manufacturer's instructions. The virus supernatant was collected 48–72 h after the transfection. Target cells were then transduced with the virus supernatant with 4 μg/ml of polybrene (Sigma-Aldrich; cat. no. H9268), and the transduced cells were selected with 0.5 μg/ml puromycin for further analysis.

**Stable cell line with PPP2R2B over-expression**. The retroviral expression plasmid with full-length PPP2R2B was constructed, and the generation of the stable cell line with PPP2R2B overexpression was described, by Tan et al.[24]. Briefly, Platinum-A cells were transfected, using Lipofectamine 2000 (Invitrogen; cat. no. 11668019), with the plasmid with PPP2R2B or the empty vector to generate retrovirus. Target cells were then infected with the virus with 4 μg/ml of polybrene (Sigma-Aldrich; cat. no. H9268) and sorted for GPF positive cells for further analysis.

**Chromatin immunoprecipitation (ChIP)-qPCR**. Procedures described previously[40] were used for the ChIP experiments. Briefly, the cells were treated with 37% formaldehyde and lysed with SDS, followed by sonication. The cell lysates were then precleared with the Protein G Agarose (Roche; cat. no. 05015952001) plus antibody against rabbit IgG (Santa Cruz Biotechnology; cat. no. sc2027), followed by overnight incubation with anti-H3K27Me3 antibody (Cell Signaling Technology; cat. no. 9733) or anti-EZH2 antibody (Active Motif; cat. no. 39901). After incubation with the antibody, magnetic Protein-G (Invitrogen; cat. no. 10004D) and a magnetic rack were used to perform the immunoprecipitation. The pull-down DNAs were then eluted, followed by overnight reverse-crosslinking at 68 °C and then DNA-cleanup with the QIAquick PCR Purification Kit (QIAGEN; cat. no. 28104). The purified immunoprecipitated DNAs and input DNAs were analyzed by qPCR, and the promoter-binding enrichments of target proteins were quantitated relatively to the input DNA. Fold changes relative to the ACTB promoter were presented. The primers that were used to amplify the promoter of PPP2R2B are listed in Supplement Table 4.

**PPP2R2B-PP2A in vitro phosphatase assay**. The lentiviral expression plasmid with FLAG fusion PPP2R2B was constructed in vector pLenti-puro (Addgene; cat. no. 39481) via EcoRV and XhoI. PPP2R2B was PCR amplified from the vector pMN GFP/IRES containing full-length PPP2R2B, constructed by Tan et al.[24]. The FLAG-tag (DNA sequence: GACTACAAAGACGATGACGACAAG) was added to the N-terminus or C-terminus of PPP2R2B through a forward or reverse PCR primer, respectively. Platinum-A cells were transfected using Lipofectamine 2000 (Invitrogen; cat. no. 11668019) with the vector containing the FLAG fusion PPP2R2B or the empty vector to generate the lentivirus. SKBR3 cells were then infected with the virus with 4 μg/ml of polybrene (Sigma-Aldrich; cat. no. H9268) and selected with the selection marker in the vector, puromycin. The selected cells were then treated with 10 μg/ml trastuzumab for 6 h. After the treatment, the cells were harvested and lysed on ice in phosphatase assay buffer (20 mM imidazole-HCl, 2 mM EDTA, 2 mM EGTA, 1 mM Na3VO4, Complete Protease Inhibitors, and 0.1% NP-40 [pH 7.0]). The lysates were then centrifuged for 10 min at 4 °C at maximum speed, followed by collection of the supernatants and measurement of the protein concentration. The same amount of lysate from each sample was incubated with ANTI-FLAG M2 affinity gel (Sigma-Aldrich; cat. no. A2220) at 4 °C for 3 h. The immuno-precipitates were then collected by centrifugation and washed with the phosphatase assay buffer for four times. The washed immuno-precipitates were each incubated with whole cell lysate from SKBR3 cells in the phosphatase assay buffer at 30 °C for 25 min. The reaction was ceased by addition of 3× SDS loading buffer and incubation at 100 °C for 4 min. After that, the gel was removed with centrifugation and the supernatants were loaded for western blot analysis.

**Bisulfite pyrosequencing**. Genomic DNA was isolated from cells using the QIAamp DNA Mini Kit (QIAGEN; cat. no. 51306) according to the user manual. The DNA was then subjected to bisulfite modification with the EZ DNA Methylation-Gold Kit (Zymo Research; cat. no. D5006) according to the

instruction manual. The regions of interest were then amplified with primer pairs specifically designed to bisulfite-modified DNA, using the PyroMark PCR Kit (QIAGEN; cat. no. 978703) according to the user manual. Each pair of primers contained one biotinylated primer and one non-biotinylated one. The biotinylated ssDNA from each PCR amplicon was then isolated and subjected to pyrosequencing using the PyroMark Q24 System (QIAGEN; cat. no. 9001514) with a specific sequencing primer. The data were analyzed with Pryomark Q24 Software 2.0. All the PCR and sequencing primers were designed with the PyroMark Assay Design SW 2.0 (QIAGEN), and are listed in Supplement Table 5.

**Animal study**. All experimental or surgical protocols were conducted after receiving the approval from Institutional Animal Care and Use Committee of Singapore (IACUC), the Agency for Science, Technology, and Research (A*STAR). Six-week-old female NCR nude mice were implanted with 17β-estradiol 60-day release pellets (Innovative Research of America; cat. no. SE-121) on the lateral side of the necks. Four days after the implantation, the mice were injected subcutaneously in both sides of the flank with $3 \times 10^6$ UACC812 or MB361 cells each, resuspended in 50 μl PBS and mixed 1:1 with matrigel (BD Biosciences; cat. no. 354248). When the tumors reached about 65 mm³, the mice were randomized into four groups, the vehicle administered, EPZ-6438 (EPZ), trastuzumab, and the combination of EPZ and trastuzumab. The randomization was conducted by equally dividing the tumor-bearing mice with similar tumor burden into each group. EPZ was formulated with 0.5% sodium carboxymethyl cellulose plus 0.1% Tween 80 in water and administered once per day via oral gavage at a dose of 250 mg/kg. Trastuzumab was formulated with its solvent (provided by Roche) and administered once per week via intraperitoneal injection at a loading dose of 30 mg/kg and subsequently, 15 mg/kg[47]. EPZ or trastuzumab was administered at a volume of 10 μl/g or 5 μl/g, respectively. Six hours after the last dose of treatment, the tumors were excised for IHC. Body weight of the mice was measured daily during the first seven days of the treatment, and then twice weekly until the end of the experiment. Tumor volume was measured two times per week with a Vernier caliper, and calculated in a formula as: $V = L \times W \times W/2$. Tumors with sizes displaying more than twice the s.d. of the mean at the point of randomization were excluded for analysis, and mice that died from unexpected illness were also excluded. Measurement of the tumor size was performed by a person blinded to the treatment groups. The Response Evaluation Criteria in Solid Tumours (RECIST) criteria were used to stratify tumors into progressive disease (PD), stable disease (SD), partial response (PR), or complete response (CR), whereby, PD: at least a 20% increase in tumor size; SD: a increase of <20% to a decrease of <30% in tumor size; PR: at least a 30% decrease in tumor size; CR: disappearance of the tumors. All mice were maintained at 21 °C ± 1, 55 to 70% humidity, and with a 12 h light/ dark cycle, from 7 am to 7 pm.

**Immunohistochemistry (IHC)**. Briefly, tumor tissues were paraffin-embedded and sectioned, followed by deparaffinization and rehydration, and retrieval of antigens with the Antigen Retriever (Sigma-Aldrich; cat. no. C9999). The tumor sections were then incubated with 3% H2O2 to block endogenous peroxidase and further blocked with 3% BSA, followed by incubation with primary antibody anti-phospho-rpS6 (Ser235/236) (cat. no. 2211), anti-phospho-4EBP1 (Ser65) (cat. no. 9451), or anti-phospho-4EBP1 (Thr37/46) (cat. no. 2855) purchased from Cell Signaling Technology. The sections were then incubated with Goat Anti-Rabbit IgG (H + L)-HRP Conjugate (Bio-Rad; cat. no. 1706515), counterstained with hematoxylin, and scaned with Axio Scan Z1 (Zeiss). One representative image from each tumor was selected for quantification, by a person blinded to the treatment group. The quantification was performed with ImageJ with a plugin color deconvolution downloadable online (https://beardatashare.bham.ac.uk/getlink/fiGsh6oznHZG4BvQ84YSetim/colour_deconvolution.zip). The brown layer deconvoluted was used to perform the quantification. Data were presented as the percentage of the stained area over the whole image. Tumors from more than five mice from each treatment group were used for the quantification.

**Clinical samples and data**. Human tissue samples were provided by Odense University Hospital (Denmark) and National University Hospital (Singapore). Studies with these materials were approved by the review board of each corresponding institution. Informed written consent was acquired from each individual who agreed to provide tissue samples for research purposes.

The cohort samples from treatment-naïve primary tumors and distant metastatic tumors that relapsed after trastuzumab treatment were from Odense University Hospital, Odense, Denmark. Approval from the Ethical Committee of Southern Denmark (Odense, Denmark) and the Danish Data Protection Agency (Copenhagen, Denmark) was granted. The RNAs of the tumors were extracted from frozen incisional biopsies using TissueLyser II (QIAGEN; cat. no. 85300), following the manufacturer's instructions. The extracted RNAs were then subjected to RT-qPCR for assessing gene expression.

The cohort treated with the neoadjuvant lapatinib-containing regimen comprises 34 stage I–III HER2+ breast cancer patients enrolled in a single-arm phase II clinical trial (ClinicalTrials.gov; Identifier: NCT01309607) and treated with neoadjuvant lapatinib plus chemotherapy (paclitaxel and carboplatin) for a total of four cycles. Tumor sizes were measured clinically before and after two or four cycles of treatment. RNAs of the baseline tumors and normal breast tissues

were reverse-transcribed into cDNAs, and the cDNAs were pre-amplified before being subjected to qPCR. The qPCR analysis was performed by a person blinded to changes of the tumor sizes.

The cohort treated with the neoadjuvant trastuzumab-containing regimen (trastuzumab and paclitaxel) has been described by Triulzi et al.[38] in which 24 patients with locally advanced HER2+ breast cancer were treated with one dose of trastuzumab and four subsequent cycles of trastuzumab and paclitaxel, preoperatively. The data is accessible in Gene Expression Omnibus (GEO) database (https://www.ncbi.nlm.nih.gov/geo/) as dataset GSE62327.

**Statistical analysis**. All in vitro experiments were repeated at least two times in technical triplicates unless stated otherwise. All measurements were obtained from distinct samples rather than measured repeatedly. All data are analyzed with GraphPad Prism version 8 (GraphPad Software; San Diego, CA) and expressed as means ± s.d., unless stated otherwise. Statistical significance was calculated using two-tailed student's $t$-test, unless stated otherwise. A $P$ value of less than 0.05 is considered statistically significant, unless stated otherwise, whereby, n.s., not significant; $*P < 0.05$, $**P < 0.01$, $***P < 0.001$, $****P < 0.0001$.

**Reporting summary**. Further information on research design is available in the Nature Research Reporting Summary linked to this article.

## Data availability
The online dataset GSE62327 is available on GEO database (https://www.ncbi.nlm.nih.gov/geo/). Source data are provided with this paper.

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

## Acknowledgements

This study was supported by the Agency for Science, Technology and Research of Singapore (A*STAR) and the Singapore Ministry of Health's National Medical Research Council Open Fund Individual Research Grants (NMRC/OFI RG/0023/2016 to Q.Y.) and National Natual Science Foundation of China (81603342 to P. W.). This study was also supported by a Cancer Science Institute of Singapore PhD Graduate Scholarship to Y.B. We thank Mei Yee Aau, Shu Yi Lau, Li Qing Lim, and Larissa Lim for assistance with the experiments.

## Author contributions

Q.Y. and S.C.L. supervised the study. Y.B. contributed to the design, performance, and interpretation of all experiments. G.O. conducted bioinformatics and contributed to the performation of statistical analyses. W.C.L. contributed to the performance of xenograft tumor collection and tumor weight measurement. P.L.L. performed CoIP. K.G. performed IHC. J.L. contributed to the performance of western blot and RT-qPCR. P.W., P.E.L., S.E., H.J.D., E.Y.T., and S.C.L. provided essential reagents, cell lines, or patient samples. H.J.D., A.W., and S.C.L. contributed to clinical data analyses and interpretation. Y.B. and Q.Y. wrote the manuscript with input from all co-authors.

## Competing interests

The authors declare no competing interests.
