## [Peer Review File · Nature Communications]

Reviewers' Comments:

Reviewer #1:

Remarks to the Author:

In the manuscript titled "EZH2-mediated PP2A inactivation confers resistance to HER2-targeted breast cancer therapy" Bao et al., describe the role of protein phosphatase 2A (PP2A) regulatory subunit PPP2R2B in anti-HER2 response. They use bioinformatic data mining to identify PPP2R2B downregulation in poor clinical outcome and resistance to HER2-therapy in a subset HER2+ breast cancers. They claim that this downregulation is result of EZH2-mediated histone modification, resulting in sustained phosphorylation of PP2A targets p70S6K and 4EB1. The finding of PP2A role in breast cancer and especially in modulation of anti-HER2 therapies is novel and interesting. Overall, this is a well executed multidisciplinary work, which clearly suggest the relationship between PPP2R2B expression and HER2 therapy responsiveness, as well as the role of EZH2 in regulation of PPP2R2B expression. However, the exact molecular mechanism of PPP2R2B in PP2A complex and in the HER2-therapy responsiveness is not yet completely proven. Are the p70S6K and 4EBP only mTOR targets affected and are other pathways than mTOR also affected? The manuscript would greatly benefit from a global phosphoproteomic analyses, for example of the BT474, SKBR3 and UACC812 with and without the lapatinib and trastuzumab treatments.

Reviewer #2:

Remarks to the Author:

In this manuscript the authors argue that PPP2R2B suppression confers resistance to anti-HER2 therapy in an EZH2-dependent manner. Analysis of patient tumor data revealed correlation between low level of PPP2R2B and resistance to anti-HER2 therapy, suggesting a potential role of PPP2R2B in sensitivity to anti-HER2 therapies. To validate the functional role of PPP2R2B in anti-HER2 therapeutic response, they compared therapeutic response to anti-HER2 therapy and the PI3K signaling pathway in BT474 (PPP2R2B high) and SKBR3 and UACC812 (PPP2R2B low) BrCa cell lines. PPP2R2B high cell line was more sensitive to anti-HER2 therapy (as seen by robust growth inhibition) compared to PPP2R2B low cell line. Overexpression of PPP2R2B in SKBR3 re-sensitized the anti-HER2-resistant SKBR3 cells to anti-HER2 agents. Conversely, suppression of PPP2R2B in BT474 (PPP2R2B high) cell line resulted in resistance and increased levels of p-p70S6K, p-rpS6, rpS6, and 4EBP1, (components of the mTOR pathway), suggesting the PPP2R2B modulates the phosphorylation of TOR pathway components that affects the response to HER2 therapy.

They then try to find the mechanism by which PPP2R2B is suppressed. Treatment with EPZ-6438 (EZH2i) led to increased PPP2R2B expression in SKBR3 and UACC812. ChIP for EZH2 and H3K27me3 show significantly higher enrichment of EZH2 and H3K27me3 at the PPP2R2B promoter region in SKBR3 cells compared to BT474 cells, indicating EZH2-mediated PPP2R2B repression. EZH2 suppression in SKBR3 abolished the enrichment. EZH2 suppression in SKBR3 enhanced growth inhibition of lapatinib and enabled lapatinib to effectively inhibit the phosphorylation of p70S6K, rpS6 and 4EBP1. Colony growth assay showed that the combination of EZH2 inhibition and anti-HER2 therapy elicits a more robust anti-cancer effect in vitro and in vivo. Then they showed PPP2R2B downregulation was associated with acquired resistance to anti-HER2 therapy. They established acquired resistance in BT474 to anti-HER2 agents. These treatment-resistant lines had lower levels of PPP2R2B and showed enriched EZH2 binding to PPP2R2B promoter site as well as enriched H3K27me3 levels. They then show clonal cellular heterogeneity that accounts for heterogeneous subclones that express different levels of PPP2R2B, which confers differential responses to anti-HER2 treatment.

The strongest part of this manuscript relates to the study of manipulating PPP2R2B or EZH2 in cell lines. The clinical correlations are variable in their strength and do not imply mechanistic relationships. Moreover, in some of these studies, the number of patient samples are small enough to undermine the identification of a single gene as differentially expressed. As such, the details and

rigor of these studies are critical to the generalizability of these findings. It is of concern that validation of anti-HER2 therapeutic response and PI3K signaling pathway was done in only one cell line (BT474). The resistance phenotype may be cell-line specific, so these studies need to be extended to other cell lines.

Overexpression of EZH2 in BT474 to see decreased PPP2R2B would strengthen Figure 4A. Currently the data only shows inhibition of EZH2 leads to increased expression of PPP2R2B. It would also be nice to see basal levels of EZH2 in BT474 and SKBR3. What if these lines don't have similar levels of EZH2?

Further evidence is necessary to validate that EZH2 is acting directly or primarily on PPP2R2B. For example, the authors should perform ChIP in an isogenic system? BT474 and SKBR3 may harbor mutations in other genes involved in the PI3K pathway or other oncogenic pathways. It would have been better to include BT474(PPP2R2B KD) vs BT474 (parental) and SKBR3 (PPP2R2B overexpression) vs SKBR3 (parental) as ChIP conditions.

The authors have not presented any biochemistry to support that PPP2R2B is the phosphatase directly responsible for the phosphorylation events noted. Have they confirmed that this subunit is involved in productive complexes and what is the effect of depleting this PP2A subunit on other PP2A complexes in the cells.

Overall, although there is some data that supports the authors' assertions, the work presented is performed in too few cell lines to assess whether this work can be generalized.

Minor issues:

The authors should include p-values for Figure 4A for UACC812.

Why did they pick BT474 and SKBR3? Why are they using SKBR3 vs UACC812 in many of the experiments?

I think clonal heterogeneity is a widely-demonstrated phenomenon and does not add much to the manuscript.

There are formatting errors for citations of the references in the text.

Reviewer #3:

Remarks to the Author:

Resistance to anti-HER2 therapies for the treatment of HER2+ breast cancer remains a significant clinical challenge. This manuscript focuses on identifying a novel mechanism of resistance to such inhibitors that involves transcriptional repression of the gene encoding PPP2R2B, a regulatory subunit of the phosphatase, PP2a. The report begins with several analyses of publicly available datasets to demonstrate that low PPP2R2B mRNA expression is associated with resistance to lapatinib in HER2+ disease and worse outcomes. Expression levels of PPP2R2B were manipulated in HER2+ breast cancer cell lines that either respond or are resistant to HER2-targeting drugs to demonstrate that altering expression impacts the response to these drugs. EZH2 is identified as a mechanism that represses PPP2R2B gene expression in resistant cells and use of an EZH2 inhibitor improves response to trastuzumab in a single mouse model and can restore sensitivity in vitro in a model of acquired resistance. Correlative changes in the AKT signaling pathway were observed and attributed to increases in PPP2R2B expression.

While this study addresses a major clinical problem and clearly demonstrates a role for EZH2 in promoting resistance to HER2-targeted therapies, it is less clear that PPP2R2B is the primary

mediator of EZH2 effects. This is due, in part, the lack of demonstration in any study that endogenous PPP2R2B protein is altered with shRNAs to PPP2R2B, that the different cell lines used in the study have a clear difference in PPP2R2B protein levels, that overexpressed PPP2R2B achieves a level that is consistent with endogenous levels in sensitive cell lines, that acquired resistance is associated with an increase in PPP2R2B protein, or that EZH2 inhibition increases the PPP2R2B protein to a level to an appreciable level that is similar to sensitive cells. As currently written, only endogenous PPP2R2B mRNA is measured throughout and it is not clear that this translates to changes in protein levels. For example, treatment with EPZ leads to a 2-3 fold increase in PPP2R2B mRNA in UACC812 cells (fig. 4A) over a background of very low expression [$<10\%$ of BT-474 at baseline (fig. 2A)]. It is not obvious that this would be sufficient to increase PPP2R2B protein levels to that needed to induce sensitivity to HER2 therapies. The lack of assessments of PPP2R2B endogenous protein changes undermines the argument that the efficacy of EZH2 inhibitors primarily function through induction of PPP2R2B to enhance HER2-targeted therapies.

In addition to this major limitation, three additional major concerns were noted. The first is the use of a single xenograft model to demonstrate the *n vivo* combined efficacy of EZH2 and HER2-targeted therapy. While these data demonstrate a remarkable combinatorial effect, the experiments should be expanded to include additional models, including a PDX model. The second revolves around the novelty of determining that EZH2 is a mediator of resistance to HER2 therapies. This was previously shown in an immune-competent mouse model and the authors of that manuscript concluded that the impact of EZH2 inhibition was mediated by changes in interferon signaling (Hirukawa, et al, 2019, Cell Reports). This paper was neither discussed nor cited. Lastly, given the considerable utilization of datasets for examining the gene expression patterns of PPP2R2B, it would seem useful to determine if EZH2 and PPP2R2B are inversely correlated in HER2+ disease, but this was not included.

Minor concerns:

- 1) Figure 3F—it is unclear what comparisons the significance markers are addressing. Are the shPPP2R2B values different than the shNC control?
- 2) Figure 4F—this experiment is lacking the control of siEZH2 without the addition of lapatinib. Is EZH2 loss sufficient to inhibit growth and the addition of lapatinib adds no benefit?
- 3) Figure 5I—it appears that the tumor morphology is different between vehicle and EPZ treated tumor. Is this true or were none similar regions of tumors evaluated for the two treatment groups. These data should be quantified with multiple animals in each treatment group.
- 4) For the KM Plotter studies, the authors should include gene expression cut-offs for each group as well as include the identifiers for the probes that were analyzed.
- 5) It is not clear why the TCGA or METABRIC datasets were not utilized.
- 6) Quantitative measures of synergy should be included for the *in vitro* drug studies such as Combination Indices.
- 7) Unless all studies using EPZ are replicated with siEZH2, a second EZH2 inhibitor should be evaluated to rule out off-target effects.
- 8) There are several grammatical/spelling errors.

Point by point response to REVIEWER COMMENTS

Reviewer #1 (Remarks to the Author):

In the manuscript titled “EZH2-mediated PP2A inactivation confers resistance to HER2-targeted breast cancer therapy” Bao et al., describe the role of protein phosphatase 2A (PP2A) regulatory subunit PPP2R2B role in anti-HER2 response. They use bioinformatic data mining to identify PPP2R2B downregulation in poor clinical outcome and resistance to HER2-therapy in a subset HER2+ breast cancers. They claim that this downregulation is result of EZH2-mediated histone modification, resulting in sustained phosphorylation of PP2A targets p70S6K and 4EB1. The finding of PP2A role in breast cancer and especially in modulation of anti-HER2 therapies is novel and interesting. Overall, this is a well executed multidisciplinary work, which clearly suggest the relationship between PPP2R2B expression and HER2 therapy responsiveness, as well as the role of EZH2 in regulation of PPP2R2B expression. However, the exact molecular mechanism of PPP2R2B in PP2A complex and in the HER2-therapy responsiveness is not yet completely proven.

Are the p70S6K and 4EBP1 only mTOR targets affected and are other pathways than mTOR also affected? The manuscript would greatly benefit from a global phosphoproteomic analyses, for example of the BT474, SKBR3 and UACC812 with or without the lapatinib and trastuzumab treatments.

Response: We agree that a global search can be helpful in identifying “all” the potential targets of PP2A. However, we took a hypothesis-driven approach and decided to focus on the pathways previously known to be associated with anti-HER2 response in breast cancer, and those previously shown to be substrates of PP2A. This approach is sufficient to lead the identification of p70S6K and 4EBP1, but not other known PP2A substrates such as AKT, MYC, etc., that are associated with resistance to anti-HER2. Importantly, in the revised manuscript, we have performed in vitro phosphatase assay and demonstrated that PPP2R2B-associated PP2A complex specifically targets p70S6K and 4EBP1 for dephosphorylation, but not AKT, ERK and MYC (Figure 3D). However, we agree that there might be some other molecules that could be affected by PP2A. We could not exclude this.

Reviewer #2 (Remarks to the Author):

In this manuscript the authors argue that PPP2R2B suppression confers resistance to anti-HER2 therapy in an EZH2-dependent manner. Analysis of patient tumor data revealed correlation between low level of PPP2R2B and resistance to anti-HER2 therapy, suggesting a potential role of PPP2R2B in sensitivity to anti-HER2 therapies. To validate the functional role of PPP2R2B in anti-HER2 therapeutic response, they compared therapeutic response to anti-HER2 therapy and the PI3K signaling pathway in BT474 (PPP2R2B high) and SKBR3 and UACC812 (PPP2R2B low) BrCa cell lines. PPP2R2B high cell line was more sensitive to anti-HER2 therapy (as seen by robust growth inhibition) compared to PPP2R2B low cell line. Overexpression of PPP2R2B in SKBR3 re-sensitized the anti-HER2-resistant SKBR3 cells to anti-HER2 agents. Conversely, suppression of PPP2R2B in BT474 (PPP2R2B high) cell line resulted in resistance and increased levels of p-p70S6K, p-rpS6, rpS6, and 4EBP1, (components of the mTOR pathway),

suggesting the *PPP2R2B* modulates the phosphorylation of TOR pathway components that affects the response to HER2 therapy.

They then try to find the mechanism by which *PPP2R2B* is suppressed. Treatment with EPZ-6438 (EZH2i) led to increased *PPP2R2B* expression in SKBR3 and UACC812. ChIP for EZH2 and H3K27me3 show significantly higher enrichment of EZH2 and H3K27me3 at the *PPP2R2B* promoter region in SKBR3 cells compared to BT474 cells, indicating EZH2-mediated *PPP2R2B* repression. EZH2 suppression in SKBR3 abolished the enrichment. EZH2 suppression in SKBR3 enhanced growth inhibition of lapatinib and enabled lapatinib to effectively inhibit the phosphorylation of p70S6 and 4EBP1. Colony growth assay showed that the combination of EZH2 inhibition and anti-HER2 therapy elicits a more robust anti-cancer effect in vitro and in vivo. Then they showed *PPP2R2B* downregulation was associated with acquired resistance to anti-HER2 therapy. They established acquired resistance in BT474 to anti-HER2 agents. These treatment-resistant lines had lower levels of *PPP2R2B* and showed enriched EZH2 binding to *PPP2R2B* promoter site as well as enriched H3K27me3 levels. They then show clonal cellular heterogeneity that accounts for heterogeneous subclones that express different levels of *PPP2R2B*, which confers differential responses to anti-HER2 treatment.

The strongest part of this manuscript relates to the study of manipulating *PPP2R2B* or EZH2 in cell lines. The clinical correlations are variable in their strength and do not imply mechanistic relationships. Moreover, in some of these studies, the number of patient samples are small enough to undermine the identification of a single gene as differentially expressed. As such, the details and rigor of these studies are critical to the generalizability of these findings. It is of concern that validation of anti-HER2 therapeutic response and PI3K signaling pathway was done in only one cell line (BT474). The resistance phenotype may be cell-line specific, so these studies need to be extended to other cell lines.

Response: We should clarify that apart from BT474, SKBR3 and UACC812, as well as BT474LR and BT474TR also were used to demonstrate the effect on PI3K pathway, including *PPP2R2B* manipulation (Figure 3A-C, fig. S2A, B, S9A, and B), EPZ sensitizes anti-HER2 (Figure 5A-D), BT474TR (Figure 7G), as well as UACC812 and MB361 in vivo model (Figure 6A-C). These data show the effect is NOT cell-line specific, and in fact, up to six cell models have been used to provide the robustness of the data.

Overexpression of EZH2 in BT474 to see decreased *PPP2R2B* would strengthen Figure 4A. Currently the data only shows inhibition of EZH2 leads to increased expression of *PPP2R2B*. It would also be nice to see basal levels of EZH2 in BT474 and SKBR3. What if these lines don't have similar levels of EZH2?

Response: Even the *PPP2R2B* is downregulated in resistant cell lines, the bulk level of EZH2 is not higher in the resistant line, and therefore, overexpression of EZH2 did not necessarily suppress *PPP2R2B* (data not shown). Instead, we show that EZH2-mediated repression of *PPP2R2B* is due to its increased recruitment to the *PPP2R2B* promoter in the resistant line compared to the sensitive line. To provide additional evidence to show EZH2 regulates *PPP2R2B*, we decided to take a different approach by using a set domain mutant EZH2, which can act as a dominant-negative EZH2 inhibitor. Indeed, the mutant EZH2, but not the wild type

EZH2, is able to induce *PPP2R2B* expression in SKBR3 cells. This added another piece of evidence showing EZH2 regulates *PPP2R2B*. The new data can be found in the revised fig. S4.

Further evidence is necessary to validate that EZH2 is acting directly or primarily on PPP2R2B. For example, the authors should perform ChIP in an isogenic system? BT474 and SKBR3 may harbor mutations in other genes involved in the PI3K pathway or other oncogenic pathways. It would have been better to include BT474(PPP2R2B KD) vs BT474 (parental) and SKBR3 (PPP2R2B overexpression) vs SKBR3 (parental) as ChIP conditions.

Response: We understand that it is probably not the best to compare the ChIP result between two different cell lines (as shown in Figure 4C). That is why we do have an isogenic condition in which the ChIP analyses in the parental BT474 and BT474LR were performed to give out a fair conclusion (Figure 7D). In this analysis, we observed increased occupancy of EZH2 and H3K27me3 in BT474LR when compared to BT474 (Figure 7D). However, we do not quite understand the isogenic conditions suggested by the reviewer. We are not sure what we expect to see in the *PPP2R2B* KD or overexpression condition because we do not believe this will affect the ChIP result of EZH2 and H3K27me3 on *PPP2R2B* promoter.

The authors have not presented any biochemistry to support that PPP2R2B is the phosphatase directly responsible for the phosphorylation events noted. Have they confirmed that this subunit is involved in productive complexes and what is the effect of depleting this PP2A subunit on other PP2A complexes in the cells.

Response: Thanks for the comments. We now have performed PPP2R2B-PP2A in vitro phosphatase assay and demonstrated the integrity of PPP2R2B-PP2A complex pull down. We further show that the immunoprecipitated PPP2R2-PP2A complex can directly and specifically dephosphorylate the p70S6K and 4EBP1 as substrates in the in vitro phosphatase assay. The new data can be found in the revised Figure 3D. We also confirmed that depleting *PPP2R2B* knockdown does not affect other PP2A subunits, and the new data can be found in revised fig. S2C.

Overall, although there is some data that supports the authors' assertions, the work presented is performed in too few cell lines to assess whether this work can be generalized.

Response: We have added another HER2-amplified cell line MB361 in the revised study for both in vitro and in vivo analysis. Together, we have SKBR3, BT474, UACC812, and MB361 models. On top of that, we also have isogenic BT474 lines acquired resistance to lapatinib (BT474LR) and trastuzumab (BT474TR). In total, we have 6 cell line models to support our conclusion. The new data can be found in revised Figure 6B, and revised fig. S8A-C.

Minor issues:

The authors should include p-values for Figure4A for UACC812.

Response: *P* values have been added in revised Figure 4A and 7E for cells treated with EPZ6438 or GSK126.

Why did they pick BT474 and SKBR3? Why are they using SKBR3 vs UACC812 in many of the experiments?

Response: BT474 and SKBR3 increase much faster than UACC812 in vitro. Due to this practical reason, more data have been acquired from BT474 and SKBR3, compared to UACC812. In fact, UACC812 is also heavily used in the study in Figure 4A, B, 5A, B, E, 6A, C, and fig S6B.

I think clonal heterogeneity is a widely-demonstrated phenomenon and does not add much to the manuscript.

Response: The main purpose of the clonal experiment in this study is to address if the acquired resistance is obtained due to pre-existing epigenetic heterogeneity or due to epigenetic adaptation in response to treatment. By showing the existence of pre-existing drug-tolerant clones, we propose that initial co-treatment with EPZ on the sensitive bulk cells might inhibit the acquisition of resistance to trastuzumab, by eliminating the pre-existing low-PPP2R2B expressing clones (Figure 8F).

There are formatting errors for citations of the references in the text.

Response: The formatting errors for citations has been corrected.

Reviewer #3 (Remarks to the Author):

Resistance to anti-HER2 therapies for the treatment of HER2+ breast cancer remains a significant clinical challenge. This manuscript focuses on identifying a novel mechanism of resistance to such inhibitors that involves transcriptional repression of the gene encoding PPP2R2B, a regulatory subunit of the phosphatase, PP2a. The report begins with several analyses of publicly available datasets to demonstrate that low PPP2R2B mRNA expression is associated with resistance to lapatinib in HER2+ disease and worse outcomes. Expression levels of PPP2R2B were manipulated in HER2+ breast cancer cell lines that either respond or are resistant to HER2-targeting drugs to demonstrate that altering expression impacts the response to these drugs. EZH2 is identified as a mechanism that represses PPP2R2B gene expression in resistant cells and use of an EZH2 inhibitor improves response to trastuzumab in a single mouse model and can restore sensitivity in vitro in a model of acquired resistance. Correlative changes in the AKT signaling pathway were observed and attributed to increases in PPP2R2B expression.

While this study addresses a major clinical problem and clearly demonstrates a role for EZH2 in promoting resistance to HER2-targeted therapies, it is less clear that PPP2R2B is the primary mediator of EZH2 effects. This is due, in part, the lack of demonstration in any study that

endogenous PPP2R2B protein is altered with shRNAs to PPP2R2B, that the different cell lines used in the study have a clear difference in PPP2R2B protein levels, that overexpressed PPP2R2B achieves a level that is consistent with endogenous levels in sensitive cell lines, that acquired resistance is associated with an increase in PPP2R2B protein, or that EZH2 inhibition increases the PPP2R2B protein to a level to an appreciable level that is similar to sensitive cells. As currently written, only endogenous PPP2R2B mRNA is measured throughout and it is not clear that this translates to changes in protein levels. For example, treatment with EPZ leads to a 2-3 fold increase in PPP2R2B mRNA in UACC812 cells (fig. 4A) over a background of very low expression [$<10\%$ of BT-474 at baseline (fig. 2A)]. It is not obvious that this would be sufficient to increase PPP2R2B protein levels to that needed to induce sensitivity to HER2 therapies. The lack of assessments of PPP2R2B endogenous protein changes undermines the argument that the efficacy of EZH2 inhibitors primarily function through induction of PPP2R2B to enhance HER2-targeted therapies.

Response: We have addressed this issue with the new PPP2R2B antibody. The endogenous PPP2R2B protein levels in various western blot experiments have been provided in the revised manuscript, including the endogenous PPP2R2B protein levels in SKBR3 and UACC812 compared to BT474, in BT474 received *PPP2R2B* knockdown, and in cells treated with EZH2 knockdown or inhibition, as in revised Figure 2A, 3F, 4B, 4E, 4F, 5C, 5D, and fig S8, A and B. The new data shows that EPZ can induce 2-3 fold induction of PPP2R2B protein as consistently with the mRNA change. Also, the overexpressed PPP2R2B in BT474 LR cells delivered a level of PPP2R2B protein slightly higher to that in the parental BT474 cells (fig S9B). Therefore, the level of overexpressed PPP2R2B is within the physiological levels.

In addition to this major limitation, three additional major concerns were noted. The first is the use of a single xenograft model to demonstrate the *n vivo* combined efficacy of EZH2 and HER2-targeted therapy. While these data demonstrate a remarkable combinatorial effect, the experiments should be expanded to include additional models, including a PDX model.

Response: We now added another HER2+ MB361 model to show the combination effect *in vivo*. The new model can be found in the revised Figure 6B. For the request to use PDX model, this is technically difficult to deliver. We have developed a number of breast cancer PDX models, but unfortunately, only TNBC seems to be successful and expandable. We have a couple of ER+, and one HER2+ PDX seemed to be successful in the first generation, but upon expansion in the subsequent passages, they all become ER or HER2 negative. In fact, we have not seen many HER2+ PDX studies in the literature, especially in the evaluation of drug response that requires PDX expansion (probably due to the same technical issues when expanding the tumors).

The second revolves around the novelty of determining that EZH2 is a mediator of resistance to HER2 therapies. This was previously shown in an immune-competent mouse model and the authors of that manuscript concluded that the impact of EZH2 inhibition was mediated by changes in interferon signaling (Hirukawa, et al, 2019, Cell Reports). This paper was neither discussed nor cited.

Response: Thanks for pointing out this study. The mice we use are immune-deficient nude mice. They do not have T cells but may still possess NK cells, but trastuzumab, which is a humanized antibody, does not bind to mouse NK receptor to elicit the ADCC effect. By using this model, we could conclude that the combined efficacy of EZH2 and HER2 inhibition is not likely due to eliciting an immune response, but rather by inhibiting the intrinsic mTOR signaling. We now have cited and discussed this study in the revised manuscript.

Lastly, given the considerable utilization of datasets for examining the gene expression patterns of PPP2R2B, it would seem useful to determine if EZH2 and PPP2R2B are inversely correlated in HER2+ disease, but this was not included.

Response: The expression level of EZH2 in BT474 (PPP2R2B high) is similar to the one in SKBR3 (PPP2R2B low) (data not shown), which prompted us to speculate that it was the occupancy of EZH2 and H3K27me3 on *PPP2R2B* promoter (Figure 4C) that causes the downregulation of *PPP2R2B* in SKBR3. We also observed no significant correlation between EZH2 and PPP2R2B mRNA levels in TCGA (data not shown).

Minor concerns:

1) Figure 3F—it is unclear what comparisons the significance markers are addressing. Are the shPPP2R2B values different than the shNC control?

Response: We performed two-way ANOVA comparing sh*PPP2R2B* and shNC, and new data can be found in revised Figure 3E and fig. S2D.

2) Figure 4F—this experiment is lacking the control of siEZH2 without the addition of lapatinib. Is EZH2 loss sufficient to inhibit growth and the addition of lapatinib adds no benefit?

Response: Data with siEZH2 alone can be found in the revised Figure 4G.

3) Figure 5I—it appears that the tumor morphology is different between vehicle and EPZ treated tumor. Is this true or were none similar regions of tumors evaluated for the two treatment groups. These data should be quantified with multiple animals in each treatment group.

Response: We do observe in general that there is less area stained in the tumors from the combination group, as compared to the untreated or the single agent. Quantification has been now provided in the revised Figure 6C.

4) For the KM Plotter studies, the authors should include gene expression cut-offs for each group as well as include the identifiers for the probes that were analyzed.

Response: The cut-offs used in the KM Plotter studies were stated in the revised legends of Figure 1A, and fig. S1B. The identifiers for the probes were present in Supplementary Table 1.

5) It is not clear why the TCGA or METABRIC datasets were not utilized.

Response: Gobo and KM plotter online analysis have been widely used for survival analysis, including our own group.

6) Quantitative measures of synergy should be included for the in vitro drug studies such as Combination Indices.

Response: In fact, in our original manuscript, we did not intend to claim the combination effect is synergistic. However, the additive values of the combinations were computed (based on the efficacies of the single agent) and shown in the revised Figure 5A and 5B, in comparison with the actual experimental combinational effects. The actual combinational effect is stronger than the computed additive effect, suggesting that there is a synergistic effect from the EZH2 and HER2 inhibition.

7) Unless all studies using EPZ are replicated with siEZH2, a second EZH2 inhibitor should be evaluated to rule out off-target effects.

Response: We performed in vitro experiments with another EZH2 inhibitor, GSK126, and the new data can be found in revised Figure 4A, 4B, 7E, and fig. S6, A and B.

8) There are several grammatical/spelling errors.

Response: We have corrected the possible spelling errors.

Reviewers' Comments:

Reviewer #1:

Remarks to the Author:

The authors have adequately addressed the concerns raised by this reviewer.

Reviewer #2:

Remarks to the Author:

The authors have addressed some of the criticisms raised in the first round of review but have mostly argued why they do not believe that the reviewer's comments need to be addressed. Although the study remains of some interest, the authors have not addressed several major issues.

1. The correlation to clinical outcome cannot be used to assert causality. The authors have elected not to address this question, which is the basis of what they argue is the clinical implication of the work.

2. The IP phosphatase assay will always allow one to say that a phosphorylated protein is dephosphorylated. Phosphatases when isolated away from their endogenous complexes will act promiscuously with phosphorylated proteins. Thus, the pull down assay is a good addition but simply does not go far enough to represent a rigorous demonstration of activity in this context.

3. The variable levels of the proteins studied in this report are concerning that this may well contribute to the trends observed. Although the authors have added an additional cell line, the variability together with the correlative data raised in point #1 make it impossible to come to the conclusions argued by the authors.

Reviewer #3:

Remarks to the Author:

The major concerns of this reviewer have been addressed. The manuscript reports the rigorous and important finding that EZH2 regulates response to HER2-targeting therapies in breast cancer, in part, by modulating the expression of PPP2R2B.

Reviewer #1 (Remarks to the Author):

The authors have adequately addressed the concerns raised by this reviewer.

Reviewer #2 (Remarks to the Author):

The authors have addressed some of the criticisms raised in the first round of review but have mostly argued why they do not believe that the reviewer's comments need to be addressed. Although the study remains of some interest, the authors have not addressed several major issues.

1. The correlation to clinical outcome cannot be used to assert causality. The authors have elected not to address this question, which is the basis of what they argue is the clinical implication of the work.

Answer: we understand that any clinical analysis of a gene behaviour to clinical outcome is always correlative in nature and not mechanistic. This is exactly why mechanistic investigations need to follow to validate the hypothesis proposed based on a correlative observation. We agree that the sample size in each cohort might not be high enough but the conclusion is based from independent analysis of different cohorts.

2. The IP phosphatase assay will always allow one to say that a phosphorylated protein is dephosphorylated. Phosphatases when isolated away from their endogenous complexes will act promiscuously with phosphorylated proteins. Thus, the pull down assay is a good addition but simply does not go far enough to represent a rigorous demonstration of activity in this context.

Answer: this is not necessarily true. Please note that the ppp2r2b pull down assay only shows specific dephosphorylation of p70S6K and 4EBP1 but not Myc, ERK and AKT, which indicates that the dephosphorylation event is specific and not promiscuously with phosphorylated protein in general. It is also important to note the result of in vitro phosphatase assay is highly consistent with the bulk western blot analysis showing ppp2r2b expression is correlated with reduced phosphorylation of p70S6K and 4EBP1 but not AKT, MYC, and ERK. We do agree that a future clinical validation of PPP2R2B in relation to mTOR and resistance to anti-HER2 will be helpful to consolidate the conclusion.

3. The variable levels of the proteins studied in this report are concerning that this may well contribute to the trends observed. Although the authors have added an additional cell line, the variability together with the correlative data raised in point #1 make it impossible to come to the conclusions argued by the authors.

Answer: the nature of the rigorous investigations including gene knockdown and overexpression in various

models including both intrinsic and acquired resistance models have clearly validated our hypothesis.

Reviewer #3 (Remarks to the Author):

The major concerns of this reviewer have been addressed. The manuscript reports the rigorous and important finding that EZH2 regulates response to HER2-targeting therapies in breast cancer, in part, by modulating the expression of PPP2R2B.